# Vortex entropy and superconducting fluctuations in ultrathin underdoped Bi$_2$Sr$_2$CaCu$_2$O$_{8+x}$ superconductor

Shuxu Hu[1], Jiabin Qiao [1,2,3] ✉, Genda Gu[4], Qi-Kun Xue [1,3,5,6] ✉ & Ding Zhang [1,3,6,7] ✉

Vortices in superconductors can help identify emergent phenomena but certain fundamental aspects of vortices, such as their entropy, remain poorly understood. Here, we study the vortex entropy in underdoped Bi$_2$Sr$_2$CaCu$_2$O$_{8+x}$ by measuring both magneto-resistivity and Nernst effect on ultrathin flakes (≤2 unit-cell). We extract the London penetration depth from the magneto-transport measurements on samples with different doping levels. It reveals that the superfluid phase stiffness $\rho_s$ scales linearly with the superconducting transition temperature $T_c$, down to the extremely underdoped case. On the same batch of ultrathin flakes, we measure the Nernst effect via on-chip thermometry. Together, we obtain the vortex entropy and find that it decays exponentially with $T_c$ or $\rho_s$. We further analyze the Nernst signal above $T_c$ in the framework of Gaussian superconducting fluctuations. The combination of electrical and thermoelectric measurements in the two-dimensional limit provides fresh insight into high temperature superconductivity.

Superconducting vortices, each made of a swirling supercurrent around a normal core, are vital entities not only for understanding crucial aspects of their host superconductors[1-4] but also for realizing novel functions[5-9]. Vortices played a central role in generating the diode effect in asymmetric superconductors[10,11]. This so-called vortex ratchet effect has been further developed for constructing a supercurrent diode[12], which can switch between dissipationless and dissipative states by the bias direction. Of late, investigating vortices are indispensable for identifying the topological nature of a superconductor via Majorana zero modes[13-17]. These advancements call for a deeper understanding of vortices. One powerful approach to meeting this goal is by applying a temperature gradient $(-\nabla_x T)$ along the superconductor[18-20] to drive a vortex flow below the superconducting transition temperature ($T_c$). The temperature gradient exerts a thermal force on the vortex: $F = -S_d \nabla_x T$, where $S_d$ is the transport

entropy per vortex. Due to the motion of vortices, a transverse electric field, i.e., $E_y = B\upsilon_x$ ($\upsilon_x$ is the velocity of vortices along the temperature gradient), is produced and can be experimentally measured. As a result, the Nernst effect, defined as $N = E_y / (-\nabla_x T)$, can probe directly the vortex dynamics. Such a scheme has been employed to extract $S_d$ by using the formula[18,19,21]: $S_d = N\Phi_0 / \rho_{xx}$, where $\Phi_0 = h/2e$ is the flux quantum and $\rho_{xx}$ is the flux-flow resistivity.

A recent study collected the data of $S_d$ from four distinct families of superconductors, ranging from Nb-SrTiO$_3$ with a rather low $T_c$ of 0.35 K to La$_{2-x}$Sr$_x$CuO$_4$ with a $T_c$ as high as 29 K[22]. Interestingly, the sheet entropy, defined as $S_d^{sheet} = S_d c$ ($c$ is the $c$-axis lattice constant) seems to be comparable among the four superconductors at the peak position of the Nernst signal. This investigation was soon extended to two other cuprate compounds: YBa$_2$Cu$_3$O$_{6+x}$ and Bi$_2$Sr$_2$CaCu$_2$O$_{8+x}$ (BSCCO), with $T_c$ around 90 K[23], and the isotropic superconductor of

[1]State Key Laboratory of Low Dimensional Quantum Physics and Department of Physics, Tsinghua University, Beijing, China. [2]Centre for Quantum Physics, Key Laboratory of Advanced Optoelectronic Quantum Architecture and Measurement, School of Physics, Beijing Institute of Technology, Beijing, China. [3]Beijing Academy of Quantum Information Sciences, Beijing, China. [4]Condensed Matter Physics and Materials Science Department, Brookhaven National Laboratory, Upton, NY, USA. [5]Southern University of Science and Technology, Shenzhen, China. [6]Frontier Science Center for Quantum Information, Beijing, China. [7]RIKEN Center for Emergent Matter Science (CEMS), Wako, Saitama, Japan. ✉e-mail: jiabinqiao@bit.edu.cn; qkxue@mail.tsinghua.edu.cn; dingzhang@mail.tsinghua.edu.cn

$K_3C_{60}$[24]. So far, these studies employed compounds in individual superconducting families with the optimal $T_c$. However, many of these unconventional superconductors have a dome-like dependence of $T_c$ as a function of doping. The doping evolution of vortex entropy is thus an outstanding problem that calls for experimental investigations.

Measuring the Nernst effect has also shed vital insights into the superconducting fluctuations in numerous superconductors[25–29] above the transition temperature ($T > T_c$). Xu et al.[25] first attributed the strongly enhanced Nernst signal above $T_c$ to superconducting fluctuations in the pseudogap regime. Of late, Cyr-Choinière et al.[26] pointed out that superconducting fluctuations only extend above $T_c$ to a narrow temperature window, which is far below the pseudogap temperature. Still, data points demarcating the onset of superconducting fluctuations remain limited in the phase diagram and the extremely underdoped regime (EUD) has not been explored.

Here, we address the doping dependence of vortex entropy in BSCCO with their thicknesses down to 1.5 unit-cell (UC). Pushing to the 2D limit allows us to address electrical and thermoelectric transport in highly uniform samples even in the EUD regime ($T_c \sim 7$ K). A comparative study of ultrathin and bulk-like samples clearly reveals the dimensionality effect in thermal activation of vortices. Studying the samples in the 2D limit allows us to extract quantitatively the London penetration depths. By combining the magneto-resistance and Nernst effect data from the same batch of samples, we obtain the vortex sheet entropy $S_d^{sheet}$ from the optimally doped (OP) to the EUD regime. Notably, $S_d^{sheet}$ dramatically plummets with decreasing doping, showing an exponential dependence on $T_c$. Furthermore, the prominent Nernst signal above $T_c$ extends to a limited range of about 40 K in the EUD case. Our data can be further compared with the theory for Gaussian superconducting fluctuations. In general, our work opens up further opportunities in addressing vortex dynamics and superconducting fluctuations in 2D superconductors.

## Results

### Sample fabrication

Cuprate superconductors in the underdoped regime often suffer from inhomogeneity and even phase separation. In order to study samples with high uniformity, we take advantage of the recent development in fabricating ultrathin BSCCO samples[30–34]. We tune the doping levels by the controlled release of oxygen, as displayed in Fig. 1a. This recipe starts with a prepatterned $SiO_2$/Si substrate (Fig. 1b). Its electrodes, for both standard transport and on-chip thermometry, are already wired to a chip carrier (Fig. 1a). In the inert atmosphere, we dry stamp the selected BSCCO flake onto the electrodes (further details are given in "Methods"). For ultrathin BSCCO flakes (1.5–2 UC thick), in contrast to thick ones, their doping level can move to the EUD regime once landed on the substrate. This is caused by out-diffusion of interstitial oxygen at room temperature[32]. After a controlled period of time (up to 30 min), we cap BSCCO with a flake of hexagonal boron nitride (h-BN) to suppress this out-diffusion and protect the sample against further degradation. The complete setup is quickly loaded into the cryostat (within 5 min) for measurements. The oxygen out-diffusion is eventually switched off by cooling the sample to low temperatures. We employ our recipe to realize a set of ultrathin BSCCO samples with their doping levels ranging from OP to EUD. Their temperature-dependent resistance curves are displayed in Fig. 1c. We define their $T_c$ from the temperature point where the resistance first reaches 1% of the normal state resistance with decreasing temperature (see details in Supplementary Note 5). This criterion is consistent with former studies on bulk superconductors[28]. Notably, superconducting transitions measured from different pairs of contacts on the same sample show nearly indistinguishable behaviors, attesting to the high uniformity.

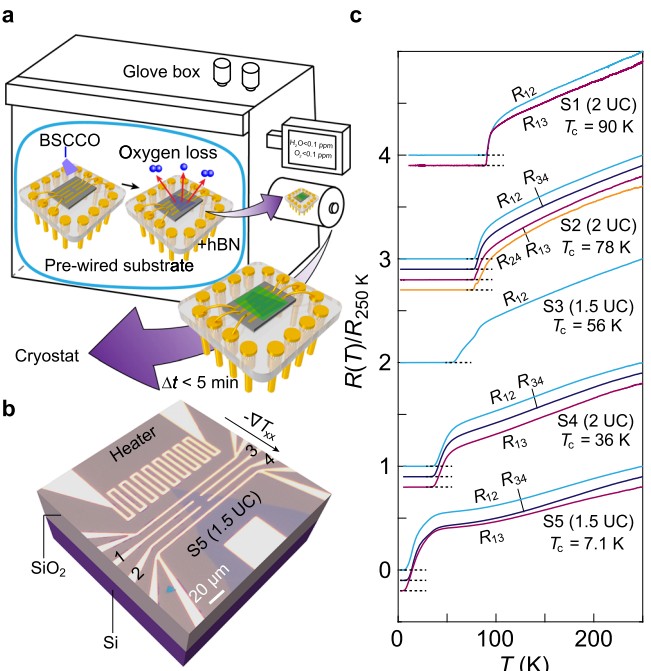

**Fig. 1 | Fabrication and characterization of ultrathin BSCCO. a** Key fabrication steps in a glovebox with inert atmosphere for realizing ultrathin BSCCO flakes with different doping levels. Purple arrows indicate the workflow. The BSCCO flake is dry-transferred onto a substrate with prepatterned electrodes that are already wired to the chip carrier. Red arrows schematically illustrate the out-diffusion of interstitial oxygen in BSCCO. In the end of the process, the complete chip is taken out of the glovebox and plugged into the measurement stick and loaded into the cryostat within 5 min. **b** Optical image of sample S5 together with electrodes for resistance and thermoelectric measurements. Numbers mark the electrodes for measuring the resistance. The arrow indicates the direction of the temperature gradient if the heater is on. **c** Normalized resistance as a function of temperature for five ultrathin samples (S1–S5). Thicknesses are indicated in the parentheses. For some samples, resistance data from different pairs of electrodes are included. The subscripts indicate the used contacts. For example, $R_{12}$ represents the resistance between electrodes 1 and 2 (indicated in (**b**)). $R_{13}$ and $R_{24}$ are mainly from the longitudinal component that mixes into the transverse resistance due to slight misalignment between the current and the Hall contacts. Curves are vertically offset for clarity. Dashed lines mark zero resistance for each curve. For each sample, $T_c$ is defined as the temperature point where the resistance reaches 1% of the normal state resistance.

### Magneto-resistivity and superfluid phase stiffness

Figure 2a–e presents the magneto-resistivity from the ultrathin samples, and Fig. 2f, g shows similar data but from bulk-like flakes. For each sample, we measure the temperature-dependent resistivity at a set of perpendicular magnetic fields (along the c-axis). We first compare ultrathin and bulk-like samples in the OP regime (Fig. 2a, f). Instead of a monotonic temperature dependence as seen in sample S1, a shoulder develops in the transition regime (60–80 K) of S6 with the applied magnetic fields. Such a shoulder was observed in early measurements of bulk BSCCO[35,36] and was later understood as an indication of current redistribution[37]. Applying a magnetic field enhances the anisotropic factor $\rho_c/\rho_{ab}$, where $\rho_c$ and $\rho_{ab}$ are out-of-plane and in-plane resistivity components. Consequently, the current gets squeezed to the layers closer to the bottom contacts (Inset of Fig. 2f). The absence of any shoulder in the data of S1 to S5 demonstrates the superiority of using ultrathin samples. By getting rid of the current redistribution problem, the measured resistivity reliably reflects the in-plane component. Comparison in the underdoped (UD) regime shows another advantage of ultrathin samples. Figure 2g shows the typical behavior of UD samples fabricated from an as-grown UD crystal. Resistivity increases with decreasing $T$ in the normal state before the superconducting

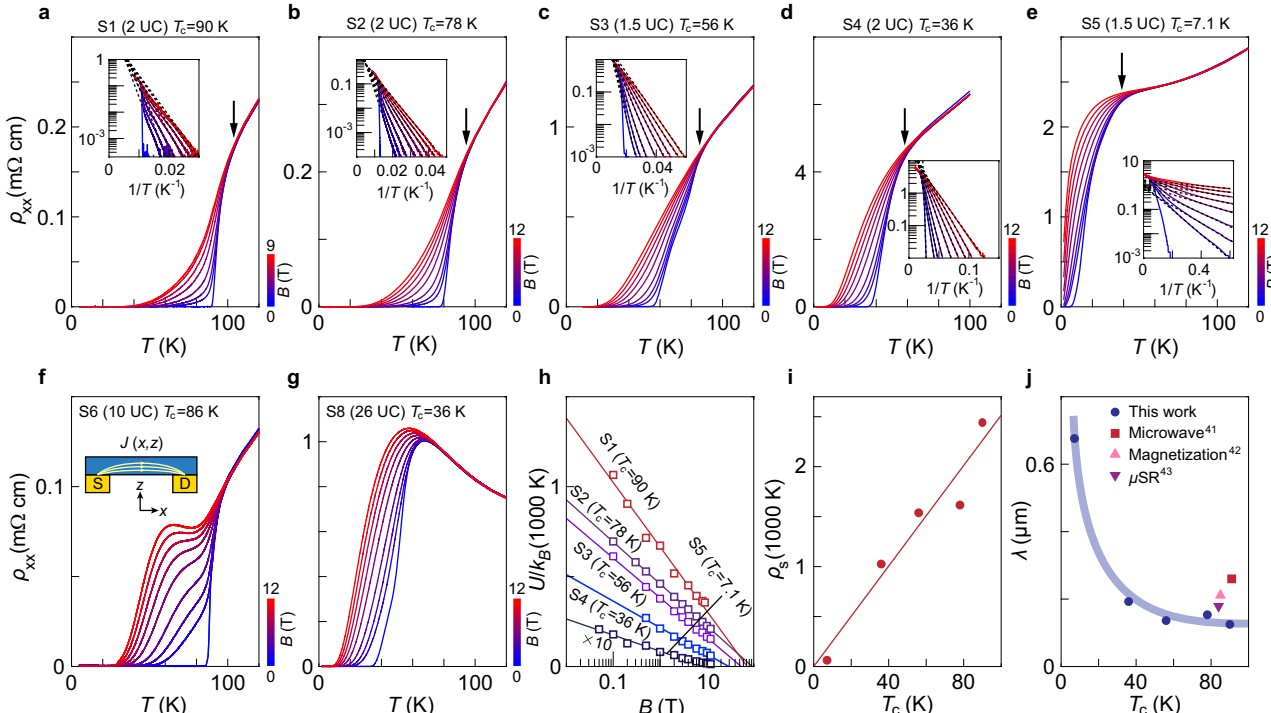

**Fig. 2 | Resistivity signals in ultrathin BSCCO with different doping levels.**
Temperature-dependent resistivity of ultrathin samples S1–S5 (**a**–**e**) and bulk-like samples S6 and S8 (**f**, **g**) at a set of perpendicular magnetic fields (*B*) (for S1, *B* = 0, 0.5, 1, 2, 4, 6, 8, 9 T; for S2–S6 and S8, *B* = 0, 0.5, 1, 2, 4, 6, 8, 10, 12 T.) Insets of (**a**–**e**) show Arrhenius plots of the data. Arrows in (**a**–**e**) mark the temperature point where the resistivity traces bifurcate. Dashed lines are linear fits. Inset in (**f**) shows the schematic drawing of the current distribution in thick BSCCO flakes. **h** Activation energy $U/k_B$ as a function of *B* for samples S1–S5. Solid lines are linear fits. **i** Superfluid phase stiffness $\rho_s$ as a function of $T_c$. Solid line is a linear fit that crosses the origin. **j** London penetration depth $\lambda$ as a function of $T_c$. Circles are depth values evaluated from the phase stiffness. Other symbols are depth values from former studies[41–43] on BSCCO single crystals. These parameters are given in Supplementary Table S1.

transition. By contrast, the ultrathin samples at the same doping and even lower doping exhibit fully metallic behavior in the normal state (Fig. 2d, e), attesting to substantially improved homogeneity.

From the magnetic field response, we can define a temperature point where the resistivity traces bifurcate (indicated by arrows in Fig. 2a–e). This temperature point indicates the onset of superconductivity. The difference between this onset and $T_c$ reflects the broad superconducting transition. Such a difference increases with decreasing doping. For samples S3 and S4, the onset temperature is roughly 20 K above $T_c$ whereas for sample S5 this temperature window is about 30 K. It indicates enhanced superconducting fluctuations in the EUD regime.

The resistivity data helps probe deeper into the vortex dynamics and the dimensionality effect. In the superconducting transition region, $\rho_{xx}$ stems from the thermally activated flux flow (TAFF) such that: $\rho_{xx} \propto \exp(-U/k_B T)$, where *U* is the activation energy of vortices. Notably, our ultrathin samples show linear dependences between *U* and *B* in the semilogarithmic plot (Fig. 2h), demonstrating that $U \propto \ln B$. This is strikingly different from the typical power law dependence $U \propto B^{-\alpha}$ of bulk samples (Supplementary Fig. 4). This qualitative difference reflects a drastic change in the vortex dynamics. Vortices in a multilayer system tend to align along the *c*-axis and form flux lines. Thermal activation deforms these flux lines[38]. For the ultrathin BSCCO, however, thermal activation is governed by the collective creeping of vortices[39] in a 2D plane. Our work therefore reveals clearly the dimensionality crossover in high-temperature cuprate superconductors from multilayer to the atomic limit.

Based on the 2D vortex creeping model, the slope of each line in Fig. 2h is directly proportional to the superfluid phase stiffness $\rho_s$ (see details in "Methods"), which is further related to the London

penetration depth $\lambda$ because $\rho_s \propto 1/\lambda^2$. A decreasing slope in Fig. 2h with lower $T_c$ therefore directly reflects reduced $\rho_s$ with decreased doping. This is summarized in Fig. 2i. Within a wide range of doping, we observe that $\rho_s$ behaves rather linearly with $T_c$, consistent with the relation[40] first proposed by Uemura et al. Our results thus extend the linear relation to the 2D limit and in the EUD situation. Apart from the linear behavior, we extract quantitatively the London penetration depth $\lambda$ (Fig. 2j). The values near OP are consistent with those determined by using bulk-sensitive techniques[41–43]. Moreover, the magnetotransport measurements yield quantitative information of $\lambda$ in the previously inaccessible EUD regime.

## Nernst effect and vortex entropy

After charactering the magneto-resistivity, we study the Nernst effect on the same batch of samples. We sweep the magnetic field at fixed temperatures and register the transverse voltage induced by a temperature gradient. Figure 3a1–a5 shows the representative traces of the Nernst signal *N*. We note that the results for S1 are in quantitative agreement with previous experiments[18] on bulk crystals at OP. The measured Nernst values in ultrathin samples are also consistent with those obtained from thicker flakes by using the same on-chip setup (Supplementary Fig. 5b1). This is different from the dichotomy in magneto-resistivity data, because thermoelectric measurements on relatively thick samples do not suffer from the current redistribution problem. In addition, the Seebeck coefficients of S1–S5 (Supplementary Fig. 6), measured essentially in the normal state at $T \gg T_c$, show behaviors consistent with previous reports (Supplementary Note 3), further confirming the reliability of our on-chip thermometry.

In Fig. 3b1–b5, we plot the temperature dependence of *N* at selected magnetic fields, showing a typical peak profile. Notably, the peak value of *N* from samples with drastically different doping levels

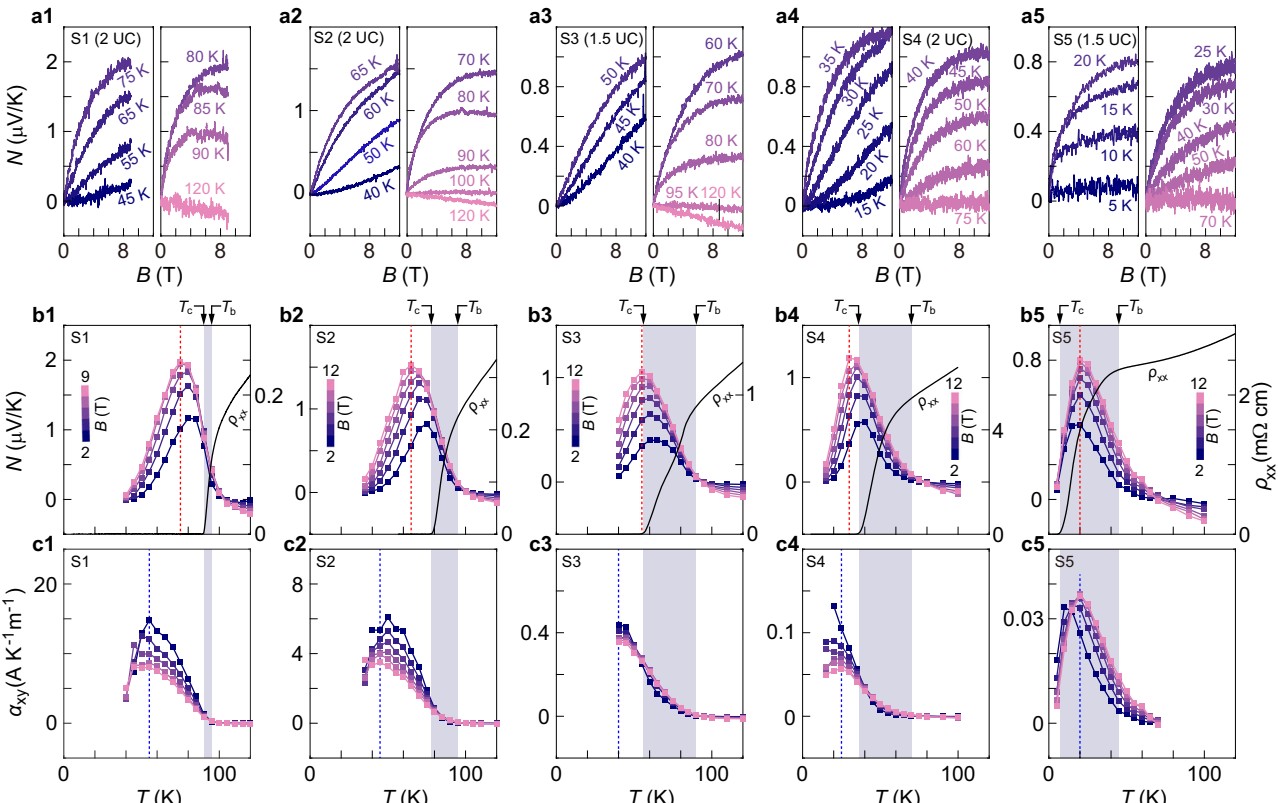

**Fig. 3 | Nernst signals and off-diagonal Peltier coefficients in ultrathin BSCCO with different doping levels. a1–a5** Nernst signals as a function of $B$ at selected temperature points for samples S1 to S5. **b1–b5** Nernst signals as a function of temperature at fixed $B$ (for S1, $B = 2, 4, 6, 8, 9$ T; for S2–S5, $B = 2, 4, 6, 8, 10, 12$ T). Red dotted lines mark peak positions of $N$ at the maximum $B$. Dashed lines mark $T_c$ defined from resistivity data. Black traces are zero-field resistivity data. Values of $N$ at each $B$ are obtained by averaging the data points on the $N(B)$ trace. For $B < 9$ T in S1 and $B < 12$ T in S2–S5, we take the average in the window of $[B − 0.5, B + 0.5]$ T. For the maximum $B$, a window of $[B − 0.5, B]$ T is chosen. **c1–c5** Off-diagonal Peltier coefficients $\alpha_{xy}$ as a function of temperature at fixed $B$ (same color coding as in (**b1–b5**)). Blue dotted lines mark peak positions of $\alpha_{xy}$ at the maximum $B$. Dashed lines mark $T_c$. Shaded regions in (**b1–b5, c1–c5**) represent the temperature window from $T_c$ to $T_b$. The latter temperature is defined as the point above which $N$ from superconducting fluctuations drops below the measurement noise level (Supplementary Note 6).

are on the same order of magnitude (a few $\mu$V/K). This echoes with the recent observation by Rischau et al.[22] There, peak values of $N$ were reported to be 4–10 $\mu$V/K among six different superconducting compounds. In contrast to the nearly invariant peak height, the peak position ($T_{N,p}$) relative $T_c$ shows a clear evolution from Fig. 3b1 to b5. $T_{N,p}$ stays below $T_c$ (dashed line) in S1 and S2 but becomes nearly coincident with $T_c$ for S3 and S4 with reduced doping. In the EUD sample S5, the peak position $T_{N,p} = 20$ K obviously exceeds $T_c = 7.1$ K but remains below the onset temperature for superconductivity.

We combine the data in Figs. 2a1–a5 and 3b1–b5 and calculate the off-diagonal Peltier coefficient $\alpha_{xy} = N/\rho_{xx}$ (Fig. 3c1–c5). We verify in Supplementary Fig. 7 that $\rho_{xx}$ and $N$ below $T_c$ are measured in the linear response regime. The temperature dependence of $\alpha_{xy}$ at a fixed $B$ also exhibits a peak, similar to that of $N$. The peak position of $\alpha_{xy}$ is at a lower temperature than that of the corresponding Nernst peak, i.e., $T_{\alpha,p} < T_{N,p}$. For samples S1–S4, $T_{\alpha,p}$ is deep below $T_c$. For S5, only data at 5 K is below $T_c$ (we discuss the contribution from superconducting fluctuations in the discussion section). The prominent signal of $N$ or $\alpha_{xy}$ below $T_c$ reflects vortex motion in the liquid state. Further lowering the temperature may drive the system into the vortex lattice phase such that $\alpha_{xy}$ diminishes. At around $T_{N,p}$ for S1–S4 and at 5 K for S5, $\rho_{xx}$ vanishes at zero magnetic field and grows linearly with $B$ across a wide field range (Supplementary Fig. 8). This trend agrees with the flux-flow (FF) behavior: $\rho_{xx} = B\Phi_0/\eta$ for a constant $\eta$. We remark that FF occurs at a relatively higher temperature and magnetic field than the regime for the TAFF, in which we extract the superfluid stiffness in the

previous section. From sample S1–S5, we observe that $\alpha_{xy}$ in this regime, which is directly linked to the vortex entropy via: $S_d = \Phi_0 \alpha_{xy}$, shows a clear drop in orders of magnitude with decreasing doping (from Fig. 3c1 to c5). It indicates a sharp decrease of vortex entropy with reduced doping. In fact, a constant entropy, i.e., a fixed ratio between $N$ and the measured $\rho_{xx}$, would require a large $N$ on the order of 100 $\mu$V/K in the EUD regime, exceeding the typical value recorded in superconductors by at least one order of magnitude[19,44].

In Fig. 4a, we compare the sheet entropy $S_d^{sheet} = S_d c$ ($c$ is the $c$-axis lattice constant) in samples S1 to S5 with those from other superconductors. For S1–S4, we extract $S_d^{sheet}$ at $T_{N,p}$ and at the highest magnetic field applied. For sample S5, the resistivity data may be contaminated by quasi-particle scattering at $T_{N,p}$ because $T_{N,p} > T_c$. We thus choose the data obtained below $T_c$ (at 5 K). As shown in Fig. 4a, a great variety of superconductors with a wide range of $T_c$ possess $S_d^{sheet}$ values clustering around $k_B$ (dashed line). At OP, our measured $S_d^{sheet}$ is in agreement with the former study on BSCCO films[23]. In the EUD regime, however, the vortex entropy of BSCCO becomes two orders of magnitude smaller than that of NbSe$_2$ with a similar $T_c$. Theoretically, the vortex entropy can be evaluated from the condensation energy of the normal core[45] (see details in "Methods"): $S_d^{cal} \sim \frac{\Phi_0^2}{\lambda^2 T}$. Based on the experimentally determined $\lambda$ and $T_c$, the theoretically expected value is $21k_B$ at OP. The theoretical value is ten times the experimental one ($2k_B$). This is different from the previous report[22], which shows 50 times difference between theory and experiment for Nb-SrTiO$_3$.

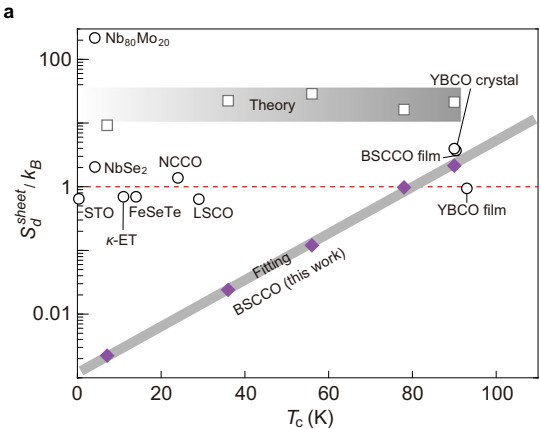

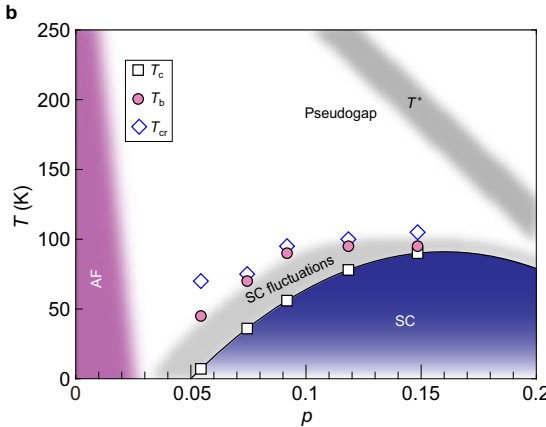

**Fig. 4 | Sheet entropy of vortices and superconducting fluctuations. a** Sheet entropy as a function of $T_c$ for a variety of superconductors. Diamonds are data from 2D BSCCO of this work. These values are evaluated at peak positions of $N$ for S1–S4 and at 5 K for S5 at the maximum magnetic field (9 T for S1, 12 T for S2–S5). Gray solid line is an exponential fit to the data. Squares indicate the theoretically evaluated entropy. Circles are entropy values at the Nernst peak positions from former studies: STO ($SrTi_{0.99}Nb_{0.01}O_3$)[22], κ-ET (κ-(ET)$_2$Cu[N(CN)$_2$]Br)[55], FeSeTe ($FeSe_{0.6}Te_{0.4}$)[56], LSCO ($La_{1.92}Sr_{0.08}CuO_4$)[57], NCCO ($Nd_{1.85}Ce_{0.15}CuO_{4+y}$)[58], YBCO single crystal and epitaxially grown film[23], epitaxially grown BSCCO film[23], NbSe$_2$[21], Nb$_{80}$Mo$_{20}$[59]. We note that the first five materials were addressed in refs. 22 and 44. Parameters used here are provided in Supplementary Table S2. **b** Summary of the superconducting fluctuation regime in the typical superconducting phase diagram of cuprates. Squares/circles/diamonds represent $T_{c,b,cr}$, respectively. The doping level $p$ is evaluated from the empirical relation $T_c = 91 \cdot \left[ 1 - 82.6(p - 0.16)^2 \right]$.

Importantly, there exists a monotonic decrease of $S_d^{sheet}$ in BSCCO with decreasing $T_c$. Such a decrease is unexpected in theory (squares in Fig. 4a). $S_d^{cal} \sim \frac{\Phi_0^2}{\lambda^2 T_c}$ should be insensitive to doping because $\lambda^2 T_c$ is a constant ($1/\lambda^2 \propto T_c$ confirmed in Fig. 2i). Intriguingly, the experimental doping dependence can be nicely captured by an exponential relation: $S_d^{sheet} \propto \exp(T_c/T_0)$, with $T_0 \sim 12$ K. It manifests itself in the semilogarithmic plot of Fig. 4a as a straight line (gray). Furthermore, our thermal activation study indicates that the superfluid density $n_s$ or $\rho_s$ has a linear relation with $T_c$. Therefore, the exponential relation of entropy vs. $T_c$ may also indicate that $S_d^{sheet} \propto \exp(n_s)$. Such a quantitative relation may guide further theoretical investigations.

## Discussion

We observe systematically reduced vortex entropy with decreasing doping. State-of-the-art scanning tunneling microscopy studies on cuprates have revealed intriguing ordering[46–48] such as pair density waves inside the vortices. Such a symmetry breaking effect, of either spin or charge, is not taken into account by the simple theoretical model that calculates the superconducting condensation energy. These orderings may lead to reduced entropy. However, the spin density wave emerges only in the EUD regime close to the antiferromagnetic insulator phase. It seems unlikely to play a major role at a doping level $p \sim 0.1$, at which the vortex entropy is already smaller than that at optimal doping. For the charge ordering, it is most prominent around the doping level of $p = 0.125$ and weakens if $p$ departs from 0.125. This non-monotonic doping dependence is incompatible with the observation of a monotonic decrease of the vortex entropy across the underdoped regime. In general, the entropy reduction seems unrelated to the well-studied spin or charge ordering.

We also discuss the implication of our experiments on the boundary of superconducting fluctuations in the phase diagram. We employ a similar protocol of Cyr-Choinière et al.[26] and determine a temperature point—$T_b$—above which the signal from superconducting fluctuations drops below our measurement noise level (marked in Fig. 3b1–b5, see details in Supplementary Note 6). Furthermore, we also obtain the temperature $T_{cr}$ where the Nernst signal changes sign. This temperature can also mark the dominance

of the contribution from quasi-particles[26]. Figure 4b summarizes $T_b$ and $T_{cr}$ over half of the superconducting dome in BSCCO. The temperature window ($T_b - T_c$) for superconducting fluctuations broadens from 5 K at OP to 40 K at EUD. Even for BSCCO in the 2D limit, the temperature regime for prominent superconducting fluctuations lies well within the pseudogap region, consistent with the finding in other cuprate bulk crystals[26].

Quantitatively, the theory of Gaussian superconducting fluctuations (GSF) yields[49,50]: $\alpha_{xy}^{GSF}/B \sim \frac{k_B e^2 (\xi_{ab}^0)^2}{6\pi \hbar^2 s} t^{-1}$, where $B \ll B_{c2}$, $\xi_{ab,c}^0$ is the in-plane coherence length, $t = T/T_c - 1$ is the reduced temperature, and $s$ is the interlayer spacing. Past studies indicate that this simple formula agree quantitatively with the experimental data from superconductors of $Nb_xSi_{1-x}$[27], $La_{1.8-x}Eu_{0.2}Sr_xCuO_4$ (Eu-LSCO)[28], $Pr_{2-x}Ce_xCuO_4$[29], etc. It is therefore interesting to check its validity in BSCCO—a superconductor with much more pronounced anisotropy. Supplementary Fig. 12 compares the Nernst coefficient data of S3 to S5 with the GSF formula. There exists reasonable agreement in a temperature window above $T_c$, indicating a dominant contribution of GSF over other factors such as quasi-particles, spin or charge order. Moreover, the fitting parameter $\xi_{ab}^0$ is consistent with the estimated coherence length based on magnetotransport. Strong deviation occurs at elevated temperatures ($T \sim 1.4 T_c$ for S3, S4, and $T \sim 4 T_c$ for S5), suggesting the dominance of quasi-particles at relatively high temperatures. This crossover again reflects the limited temperature region for superconducting fluctuations.

In summary, we carry out an extensive transport study of BSCCO in the 2D limit and from OP to EUD. The simultaneously measured magneto-resistivity and Nernst effect allow us to extract key physical parameters such as the superfluid phase stiffness/London penetration depth, the vortex entropy, and the temperature window for apparent superconducting fluctuations. While the superfluid phase stiffness varies linearly with doping, the vortex entropy decreases exponentially at lower $T_c$. The Nernst signal covering half of the superconducting dome also helps settle the long-standing controversy over superconducting fluctuations in bismuth-based high-$T_c$ superconductors. Probing electrical and thermoelectric properties in ultrathin BSCCO sets a paradigm for gaining a comprehensive and decisive understanding of emergent properties of 2D superconductivity.

## Methods

### Sample fabrication

To standardize the on-chip thermometry, we patterned a complete 4-inch silicon wafer with 285 nm thick $SiO_2$ by using photolithography and electron beam evaporation of metals (Ti/Au: 5 nm/35 nm). The wafer was then diced into rectangular substrates $10 \times 4$ mm$^2$. Each substrate hosts the same design of electrodes for both resistivity measurements and on-chip thermometry[51]. The inset in Supplementary Fig. 1a illustrates the configuration of electrodes on one such substrate. A meandering line on the top served as the heater. Two metal strips with a width of 2 μm were placed next to the heater as local thermometers (each of them has four leads). They were also used as source and drain in the resistivity measurements of the sample. Four additional electrodes were placed in between the thermometers for registering the longitudinal and transverse voltages of the sample. Prior to the stamping of BSCCO, each substrate was glued to a chip carrier with its electrodes electrically wired to the pins.

Single crystals of BSCCO were grown by the traveling floating zone method. Exfoliation and dry-transfer of BSCCO were then carried out in a glovebox with Ar atmosphere ($H_2O < 0.1$ ppm, $O_2 < 0.1$ ppm)[34]. We realized ultrathin samples from either the optimally doped (S2–S5) or overdoped (S1) single crystals. Each sample is with a fixed doping level realized by controlled oxygen release and subsequent quenching at low temperatures. In principle, the release of oxygen can be reactivated by warming up the sample to room temperature again. However, we avoided tuning the doping level this way because such a process is often accompanied by enhanced inhomogeneity, presumably because the BSCCO flakes were covered by h-BN and oxygen could only leak out from the side. The relatively thick BSCCO samples were fabricated from either the optimally doped (S6, S7) or underdoped (S8, S9) single crystals.

### Thickness characterization

We employed the atomic force microscope in the contact mode to determine the thicknesses (S1–S5) after the transport measurements. For a better characterization, we peeled off the h-BN capping layers from the ultrathin samples. Supplementary Fig. 2 shows the representative results of S2, S3, and S5. These three samples were determined to be 1.5 UC or 2 UC thick. We note that the apparent height is slightly larger than the expected thickness, presumably due to the different work functions[30].

### Transport measurements

The magnetotransport and thermoelectric measurements were carried out in two closed-cycle cryogenic systems (base temperature 1.55 K) equipped with superconducting magnets (9 T and 12 T). Electrical resistances were measured in a four-terminal configuration by using the standard lock-in technique (1 μA at 3.777 or 7.777 Hz).

For thermoelectric measurements, the calibration of the thermometers was carried out in the absence of BSCCO flakes. Specifically, we first measured the temperature-dependent resistances of the two thermometers (Th1 and Th2) in the isothermal situation (Supplementary Fig. 1a). We then passed an AC current $I_{ac}$ (1–8 mA, $\omega/2\pi = 3.777$ Hz) through the local heater (Fig. 1b). This heating current gave rise to a temperature gradient oscillating between zero and $\delta T_{1-2}$ at a frequency of $2\omega$ (with a phase delay of $\pi/2$) across the two thermometers. Experimentally, we measured the variations in resistance of the two thermometers, $\Delta R_{1,2} = \Delta V_{1,2}/I_{1,2}$, where $I_{1,2}$ were the DC current (100 μA) injected into them and $\Delta V_{1,2}$ were the resulting AC voltages in the two thermometers. The local temperature variation at Th1, 2 with a frequency of $2\omega$ was calculated by using $\Delta T_{1,2} = \frac{\Delta R_{1,2}}{dR_{1,2}(T)/dT}$. Supplementary Fig. 2b shows the measured temperature difference: $\delta T_{1-2} = \Delta T_1 - \Delta T_2$ at different heating currents. The reproducibility of

the calibration was guaranteed by measuring various sets of thermometers on multiple substrates with exactly the same geometry. We note that conventional on-chip thermometry loses sensitivity at sub-10 K regime due to the saturation of resistance. We have recently overcome this bottleneck and extended our thermometry down to 1 K by utilizing the Kondo effect[51].

For the thermoelectricity measurements of BSCCO, we chose a heating current that optimized the signal-to-noise ratio and made sure the thermometry was in the linear response regime (Supplementary Fig. 1). Both the longitudinal and transverse thermal voltages ($\delta V_{xx}$ and $\delta V_{xy}$) were recorded by the lock-in amplifiers. For measuring the Nernst effect, the amplitude of $I_{ac}$ at different temperatures was adjusted to keep the temperature difference between the two longitudinal contacts $\delta T_{xx}$ to be around 30 mK. Here the two longitudinal contacts have a separation of $l_{xx} = 8$μm. Seebeck and Nernst signals were obtained via $S = -\delta V_{xx}/\delta T_{xx}$ and $N = \delta V_{xy}/\delta T_{xy}$, where $\delta T_{xy} = \delta T_{xx} \cdot W/l_{xx}$ and $W$ was the width of the sample. We obtained the Nernst signals $N(T)$ by anti-symmetrizing the raw data to remove the possible Seebeck contribution mixed into the signal due to slight misalignment of contacts. This is realized via: $N(B) = \left[ N_{raw}^{+}(B) - N_{raw}^{-}(-B) \right]/2$, where $N_{raw}^{+}(B)$ and $N_{raw}^{-}(B)$ are traces taken at positive and negative magnetic fields, respectively.

### Thermally activated behavior of vortices

In the thermally activated flux-flow regime, the resistivity at a fixed magnetic field $B$ follows: $\rho(B,T) = \rho_0(B)e^{-\frac{U(B)}{k_B T}}$. In the Feigelman–Geshkenbein–Larkin model[39] that considers 2D collective vortex creeping, the activation energy $U(B)$ follows a logarithmic dependence on the magnetic field: $U(B) = \frac{\Phi_0^2 d}{64\mu_0 \pi^2 \lambda^2} \ln(B_0/B)$, where $d$ is the thickness of the sample. We can extract $\lambda$ by using $\lambda = \frac{\Phi_0}{8\pi} \sqrt{\frac{d}{\mu_0(-dU/d(\ln B))}}$. We further estimate the 2D superfluid phase stiffness following the formula: $\rho_s = \frac{\hbar^2 d}{4\mu_0 k_B e^2 \lambda^2}$. We point out that another model which considers the motion of thermally activated vortex-antivortex pairs[52] in 2D also gives rise to logarithmic dependence of $U(B)$. There, too, the activation energy is proportional to $1/\lambda^2$ such that the linear doping dependence of $\rho_s$ would not be affected.

### Nernst effect due to vortex flow

In deriving the formula for transport entropy per vortex $S_d = N\Phi_0/\rho_{xx}$, it is often assumed[18,19,21] that the thermal force on a vortex is balanced by viscosity such that $-S_d \nabla_x T = \eta v_x$, where $\eta$ is the viscosity coefficient. The Nernst signal is therefore:

$$N = BS_d/\eta. \tag{1}$$

However, the drifting vortex can experience the Magnus force. Consequently, it is the total force—the sum of thermal force and Magnus force—that becomes balanced by the viscous force. H.-C. Ri et al.[53] considered both the above-mentioned effect and the Hall effect of unbound quasi-particles. They obtained the following equation for the Nernst signal:

$$N = \frac{S_d \rho_{xx}}{\Phi_0} + S_n \frac{\rho_{xx}}{\rho_n} (\tan \theta_{QP} - \tan \theta_V), \tag{2}$$

where $S_n$ is the Seebeck coefficient in the normal state, $\rho_n$ is the normal state resistivity, $\theta_{QP}$ and $\theta_V$ are the Hall angles for vortices and quasi-particles, respectively. Equation (2) indicates that there exists an additional term that may contribute to the Nernst signal. We note that the two Hall angles are equal in the Bardeen-Stephen model[53] such that the second term is zero. Furthermore, the second term has a different temperature dependence than the first one in

Eq. (2). If this term has a noticeable contribution, the temperature-dependent Nernst effect should exhibit a shoulder instead of a single peak[53]. Experimentally, we observe no such a shoulder in our data (Fig. 3b1–b5). We therefore neglect the second term in evaluating the vortex entropy.

## Theoretical calculation of vortex entropy

Sergeev, Reizer, and Mitin derived the following formula for the vortex entropy[45]:

$$S_d = -\pi\xi^2 \frac{\partial}{\partial T} \frac{B_c^2}{2\mu_0},\tag{3}$$

where $\mu_0$ is the magnetic vacuum permeability. This formula considers the core energy of the vortex and neglects the contribution from the surrounding supercurrent. They argued that the supercurrent does not transport entropy. Following this expression, Rischau et al.[22] derived that:

$$S_d = -\frac{\Phi_0}{2\mu_0 \ln\kappa} \frac{\partial B_{c1}}{\partial T}.\tag{4}$$

Here $B_{c1}$ is the lower critical magnetic field and $\kappa = \lambda_0/\xi_0$ is the Ginzburg–Landau parameter. In deriving Eq. (4), they used the relations[54]: $B_{c2} = B_c \cdot \sqrt{2}\kappa$, $B_{c1} = B_c \cdot \ln\kappa/\left(\sqrt{2}\kappa\right)$, and $B_{c2} = \Phi_0/\left(2\pi\xi^2\right)$. In order to compare the theoretical value with our data at different doping levels, we aim at expressing $S_d$ by physical quantities that can be directly determined by our experiment (apart from physical constants). To do so, we input the relation $B_{c1} = B_{c2} \cdot \ln\kappa/(2\kappa^2)$ back to Eq. (4) and obtain:

$$S_d = -\frac{\Phi_0}{4\mu_0\kappa^2} \frac{\partial B_{c2}}{\partial T}.\tag{5}$$

By further employing $B_{c2} = B_{c2}(0)(1 - T/T_c)$ and $B_{c2}(0) = \Phi_0/\left(2\pi\xi_0^2\right)$, we derive that:

$$S_d = \frac{\Phi_0^2}{8\pi\mu_0\lambda_0^2 T_c}.\tag{6}$$

In Eq. (6), both the London penetration depth and the superconducting transition temperature are experimentally measurable quantities.

## Data availability

The data generated in this study have been deposited in: https://doi.org/10.6084/m9.figshare.24581010. All other data that support the plots within this paper are available from the corresponding author upon request.

## Code availability

The computer code used for data analysis is available upon request from the corresponding author.

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

## Acknowledgements

The authors thank fruitful discussions with Haiwen Liu and Zhu'an Xu. This work is financially supported by the Ministry of Science and Technology of China (2022YFA1403103) and the National Natural Science Foundation of China (Grants Nos. 12361141820, 12274249, 12204045, 52388201, and 52011530393). J.Q. was supported by the Beijing Institute of Technology Research Fund Program for Young Scholars. Work at Brookhaven is supported by the Office of Basic Energy Sciences, Division of Materials Sciences and Engineering, U.S. Department of Energy under Contract No. DE-SC0012704.

## Author contributions

S.H. and J.Q. initiated the project, fabricated the samples, and carried out transport measurements. G.G. grew the single crystals. S.H., J.Q., and D.Z. analyzed the data and wrote the paper with input from Q.-K.X.

## Competing interests

The authors declare no competing interests.
