## [Peer Review File · Nature Communications]

Vortex entropy and superconducting fluctuations in ultrathin underdoped $\text{Bi}_2\text{Sr}_2\text{CaCu}_2\text{O}_{8+x}$ superconductorREVIEWER COMMENTS

Reviewer #1 (Remarks to the Author):

Hu *et al.* study the doping dependence of the transport entropy per vortex per unit length in a cuprate superconductor of BSCCO by measuring the magneto-resistivity and Nernst effect. To control the doping level while keeping the homogeneity of the sample, the authors utilized out-diffusion of oxygen from mechanically exfoliated ultrathin samples. Using the on-chip setup, the magneto-resistivity and Nernst effect were measured successfully for the ultrathin samples with different doping levels. It is found that with decreasing the doping level, the vortex transport entropy decreases monotonically from values close to $k_B \ln 2$ in the optimally doped regime to values two orders of magnitude smaller in the extremely underdoped regime, associated with the decrease in the superfluid density and transition temperature. The authors attribute the suppression of the vortex entropy to the emergence of the charge and spin ordering expected in the underdoped regime.

Measuring the thermoelectric effect in ultrathin samples is a challenging experiment and the reported data are probably reliable. The proposed scenario that the suppression of the vortex entropy originates from the emergence of charge and spin ordering in the underdoped regime looks interesting. However, I do not find clear experimental evidence to support their assertion. It is known that in the underdoped regime of cuprate superconductors, the charge ordering appears locally around quarter filling ($p = 0.125$), as reported in Ref. 33. However, the vortex entropy shown in Fig. 4 exhibits a monotonic decrease with decreasing the doping level. I would like to ask the authors if they observe the charge and spin ordering in their samples. Also, I would ask if they confirm that the vortex entropy takes values close to $k_B \ln 2$ in the overdoped regime, instead of showing a decrease. It is necessary for the authors to present data that show directly the correlation between the vortex entropy and the degree of order in the charge and spin ordered states, thus demonstrating that the suppression of the vortex entropy is indeed caused by the charge and spin ordering.

As mentioned above, the present manuscript lacks experimental evidence to support their claim and hence is not suitable for publication in Nature Communications. However, the observed suppression of vortex entropy with decreasing the superfluid density is interesting for the specialists. Therefore, it should be considered for publication in a specialized journal if the authors address the following points adequately.

[1] In the sample S5, a peak position (T_{NP}) of the Nernst signal is well above $T_c = 5.4$ K. Then, the authors explained that the peak is caused by enhanced vortex fluctuations. This means that the vortices exist at temperatures up to 50 K. However, I have a serious concern about this interpretation. The authors mention in the introduction as: "As a result, the Nernst effect probes directly the vortex dynamics". I consider that it is not correct exactly. The Nernst effect arises from three physical origins: (i) fluctuations of the phase in the superconducting order parameter, i.e., vortices, (ii) fluctuations of the amplitude in the superconducting order parameter, which cause the paraconductivity in the electric conductivity, and (iii) quasiparticles. The authors seem to ignore the contribution of (ii). Note that in Ref. 33, Cyr-Choinière *et al.* used the term "superconducting fluctuations", instead of "vortex fluctuations". I suggest the authors to discuss their Nernst signals on the basis of Ref. 19 in your manuscript.

[2] The authors obtained theoretical values of the vortex transport entropy using the equation (1), which is converted from a theoretical expression given by Sergeev *et al.* in Ref. 23. However, the equation (1) is probably incorrect since the authors ignore the temperature dependence of the coherent length. Thus, the resulting values of entropy in Fig. 4 should be reconsidered. Furthermore, in Ref. 24 Rischau *et al.* pointed out that the expression given by Sergeev *et al.* leads to much larger values than experimental values ($\sim k_B \ln 2$) obtained by Rischau *et al.* The authors should explain why the equation (1) is valid.

[3] I cannot understand why the authors were able to measure the temperature difference between two thermometers below

20 K because the resistivity of the thermometer shows a upturn. The authors applied an ac current to a heater with a frequency of ω , producing 2ω oscillation in temperature with an amplitude of ΔT . ΔT is defined as deviation from the bath temperature T_{bath} of the cryostat. Seemingly, the authors used the value of T_{bath} to obtain the temperature difference. However, generally the minimum temperature during the temperature oscillation is larger than T_{bath} as shown in RSI 91, 095112 (2020) by Wu *et al.*

[4] The authors mention that the vortex flow in the linear I-V regime is "flux flow", where the pinning effect is absent. However, the observed Arrhenius law in the resistivity data clearly shows that the vortex flow is the thermally assisted flux flow (TAFF), where the pinning is effective [See Rev. Mod. Phys. 66, 1125 (1994)]. Thus, the authors should use the word TAFF. The discussion on the vortex entropy stays essentially unchanged whether the flow is flux flow or TAFF.

[5] Line 183 on page 10, Fig. 3c -> Fig 3b

Reviewer #2 (Remarks to the Author):

The paper by Hu *et al.* reports a Nernst effect study of the vortex entropy in BSCC superconductor. Here the authors show a decreasing entropy from optimal doping to underdoped regime in contrast to a universal value $k_B \ln 2$ as reported in ref.[24], and thus claim the breakdown of universal vortex entropy in underdoped BSCC superconductor. However, it seems that the authors don't catch the main point about vortex entropy and are less rigorous to claim such a conclusion in this manuscript. Ref.[24] focuses on the study of Nernst effect below T_c , not above T_c , due to the different mechanism. Below T_c the vortex Nernst response dominates, but above T_c , the Nernst signal from the superconducting fluctuation strongly displays the material-dependent, and the amplitude of the Nernst signal over 6 orders of magnitude is not rare. Ref.[K. Behnia and H. Aubin, Rep. Prog. Phys. 79, 046502 (2016)]. In this paper the underdoped sample is $T_c \sim 5$ K, the comparable Nernst signal may be from the superconducting fluctuation not vortex. The obtained entropy in the normal state not superconducting state is not available to the equation in the paper. Thus it is not wonder that the underdoped sample shows different behavior. For these reasons, I do not recommend the publication of this paper in NC.

In addition, it is unbelievable that the vortex fluctuation (fig.5 in SI) in EUD sample can be observed in such high temperature above T_c . Is still there a vortex liquid phase? Furthermore, the entropy density in the vortex may be different from the case of normal state.

Noted that ρ_{xx} of Sample S4 in fig.2-a4 displays the largest value below T_c , almost twice larger than that of S5, but the peak of Nernst signal of S3, S4, and S5 is almost same in fig.2 c3-c5. Why the S_d (sheet) of sample a4 in fig 3b does not be the minimum?

Reviewer #3 (Remarks to the Author):

In their manuscript, Hu and coauthors report on measurements of the Nernst effect and of the magnetoresistance in ultrathin flakes (and thicker samples) of BSCCO in a range of dopings, from optimally doped (OD) to strongly underdoped (EUD). They focus on using this data to extract the vortex entropy by taking a ratio of Nernst effect and the resistivity and find a very strong reduction of this ratio, by several orders of magnitude, when going from OD to EUD. They relate this to recent results by Behnia's group (Rischau *et al.* PRL2021) who had suggested a univesal bound (and in that paper, an upper bound was postulated) on the entropy per vortex per layer of $\ln(2)k_B$.

Upon first reading the abstract, this paper seemed to me like a clear case for rejection - after all the upper bound postulated by Richau *et al.* was supposed to be an upper bound (and was problematic in itself, see below), so seeing values below that should not come as a surprise, could

have many reasons (e.g. residual non-flux-flow resistivity), and does appear in materials that hit the "upper bound" when looking at other points in the B-T space. I still believe that in terms of the "high-level analysis" the presentation in this paper cannot remain as it is, and I could certainly not recommend publication in its current form. However, looking at the quality of the data itself, how it was taken, and the "low-level analysis", this paper is probably the best and most careful measurement of the Nernst effect in a superconductor I have seen so far.

So if the authors are willing to make the effort to put their work into a proper context, and remove the misleading claims and incorrect generalizations that currently plague the work, I believe the paper could be an extremely valuable contribution to the study of the Nernst effect in superconductors - a field of study which is both still very badly understood (so experiments are important), and very controversial (especially above T_c , and this indeed was caused by the initial rather careless presentation of results by Ong's group). It is therefore very crucial that the authors are very honest and precise in their statements. In such a case however, I could certainly see how this could become a work that should be published in this journal. In the following I will try to provide the information and comments necessary so that the authors can make the necessary improvements to bring the work to this level.

Let me provide a bit of context: In Rischau et al. PRL2021 the authors made two claims. The first is that, despite otherwise extremely different parameters (e.g. gap and T_c , lattice constant etc.), the peak (in B-T space) Nernst signal N in numerous superconductors lies in a range somewhere between 1 and 10 $\mu\text{V}/\text{K}$. Although this claim does not entirely hold up, - there are some reports in organic κ -(BEDT-TTF) $2\text{X}2$ samples which rather range in the tens of nV/K (<https://www.nature.com/articles/nature06182> and <https://www.nature.com/articles/srep03390>), although one may question those given other measurements in the same material that report higher values ([https://doi.org/10.1016/0921-4534\(96\)00243-2](https://doi.org/10.1016/0921-4534(96)00243-2)) - it does still seem to be strikingly general. In a later work by Behnia (J. Phys.: Condens. Matter 35 (2023) 074003, also cited - I will refer to it as Behnia JPCM) this value is related to a viscosity-to-entropy density ratio, and this is certainly a topic that deserves further exploration.

The second claim in that work is that when dividing the Nernst signal at its peak by the resistivity at the same field and temperature, ρ , a universal bound of $\ln(2)k_B$ per vortex per layer (in that case the "layer" is taken as one lattice constant in the field direction) is not exceeded (and the values reported in that work seem to hit that limit to two significant digits).

The latter claim had two problems. First of all, if the ratio of N/ρ gives access to the entropy S , why should one consider its value at the peak of N ? Instead, one should consider the peak of N/ρ itself. As one can see for example in work by Li et al (Chin. Phys. B Vol. 29, No. 8 (2020) 087402), also cited by the authors, this ratio can actually be much larger at lower fields than it is at the peak of N . If one wants to make general statements about vortex entropy, there is no reason to limit those studies to the peak of N , which is ultimately an experimental quantity (whereas the ratio of N/ρ gives α_{xy} , which is more intrinsic). Second, there are actually a number of materials where this "universal bound" of $\ln(2)k_B$ is exceeded, and in Behnia JPCM2023 those works are indeed mentioned. The value in most of these is still "around k_B " (and there is some uncertainty of what constitutes the length-scale of one layer).

However, importantly, in Niobium-based superconductors, the entropy exceeds $200k_B$! (see section "Historical Nernst data on superconducting niobium and its alloys" and refs 56 and 57 in that paper. So I one can clearly say that the "universal bound" for S postulated by Rischau et al. simply didn't hold up to further scrutiny, and the current manuscript submitted here is a further confirmation that $\ln(2)k_B$ is not a valid upper bound. As for a lower bound, that was not postulated there (although in Behnia JPCM there is a brief speculation, but it is enough to take any data at e.g. higher B-field, see for example Li et al, or other temperatures, to see that this does not hold up experimentally - although it is of course not clear in those other works if some non-flux-flow resistivity also comes in).

So certainly the authors cannot say, as they do in title/abstract, that theirs is an observation of a breakdown of some universal vortex entropy value. As mentioned above, the only thing that does seem somewhat universal is the peak value of N itself, and there the values that the authors find

are between 0.8 μ V/K and 2 μ V/K, so pretty much within the expected range (again, surprising given the very large change in T_c , so an interesting result in itself).

Nevertheless I believe that the authors' data is very valuable, and I will briefly mention what I find is very carefully done, in contrast to many other measurements, including recent measurements, of the Nernst effect in superconductors: There is a ρ curve associated with every N curve, the parameter range is exhaustive, linearity is checked, it is convincingly argued how and where ρ is of flux-flow type, the thermal gradients used are very small, the geometry is carefully chosen, and importantly (but unfortunately not the case in many papers), all the details of the measurement are given.

However there are a number of important points that the authors need to address (roughly in order of appearance in the paper, all are important):

1) As mentioned the title/abstract cannot stand like this. What they show is a careful and exhaustive study of the Nernst effect, Magnetoresistance and their ratio in a wide range of parameters (doping AND thin vs thick) in an underdoped cuprate. It is an important contribution to the study of vortex entropy, but it is NOT a breakdown of a universal value.

2) I. 49 What the authors ignore here is that in addition to the thermal force term there is also a Magnus force term (see Krasnov and Logvenov Physica C 274 (1997) 286-294 for theory, and [https://doi.org/10.1016/0921-4534\(96\)00243-2](https://doi.org/10.1016/0921-4534(96)00243-2) for experiments). The authors need to consider that force as well, and either take it into account or argue why it is small and can be ignored

3) Fig.1 , it is not clear how ρ_{13} or ρ_{24} are extracted, given that they are transverse to the bias current. In general there should be more details on the 4-point measurement beyond what is given (on how geometry is factored in). Also, it is not clear exactly what the measured zero value is in this plot. Is any kind of background subtracted? If so, that should be stated clearly.

4) How is T_c really defined? What does "reaches zero" mean? Reaches the signal-to-noise level? The authors should be very explicit here and also make it clear if there is any change depending on the method chosen to determine T_c (is a threshold or ρ_{100K} used? or an extrapolation of the curve?) Also, many authors rather chose the downturn or midpoint of the transition as their T_c (see also below), the author should comment on their choice.

5) The authors should plot (SI is fine) $T_c(B)$, comment on how that compares to other measurements in BSCCO and if the field-dependence of N maps to the field-dependence of $T_c(B)$ (i.e. when plotting N/B in B-T space, do the contour lines of N/B follow $T_c(B)$)Such contour plots, both for N itself and for N/B, should be included also.

6) The authors should comment on the large width of the transition even at $B=0$. How does it compare to the bulk-like samples (the $\rho(B,T)$ curves should also be shown there!)? Do they think the width is related to the low dimensionality, or rather to inhomogeneity? Does BKT physics come in? That should be mentioned.

7) I. 125 and others: Cyr-Chronière et al. do NOT claim that the onset of a B-dependent N/BT signals the onset of _vortex_ fluctuations. It is important to be precise and clear here, all statements related to that need to be changed. In the cited PRB, Cyr-Chronière et al. actually remain agnostic to whether the B-dependent N/BT they see can be explained by Gaussian fluctuations or not, all they claim is that the B-dependence of this signal (i.e. a non-linear dependence of N on B) is the signal of a Nernst effect that is related to superconducting fluctuations, rather than being purely quasiparticle-driven. Furthermore, the "onset" of something is of course just a measure of the signal to noise of the experiment! So the authors need to be clear on how they actually define the point they choose - or they better decide not to choose one specific temperature, but just to show how $d(N/B)/dB$ actually changes vs temperature!

8) In this context, the authors should make the effort and compare their results above T_c (and here they should consider different possible definitions of T_c , including onset or midpoint) to the theory of Gaussian fluctuations. This is still a open and hot debate, and as the data presented here

if of high quality, it would be important to consider this data in this context. The authors should be aware that the original work on the Nernst effect resulting from Gaussian fluctuations has by now been extended quite a bit (to also include nonlinearities), see Glatz et al. Phys. Rev. B 102, 174507 (<https://journals.aps.org/prb/abstract/10.1103/PhysRevB.102.174507>) for a recent work that also summarizes previous works. This is an important point.

9) I.131 How does being in a linear regime imply that vortex pinning is negligible (cf also Supplemtar Note 3)? Sure, if all pinning centers had exactly the same pinning force I would expect a sharp threshold, but given that pinning centers are probably widely distributed in their pinning force, that is not the case. A more careful discussion is needed here. It is almost impossible that there is no pinning in the underdoped samples, given the intrinsic disorder that has to exist there. It is good however that the authors have explicitly tested for linearity here!

10) In this context, how does $\rho(B)$ compare with $\rho(\text{normal})$ when plotting it as $\rho(B/Bc2)$ - if the authors have access to $Bc2$ ($Hc2...$) either from the extrapolation mentioned above or from literature if available?

11) I.134 "typically evaluated" is not really the case, it is still only a few works. In any case and this is a VERY important point: The authors should, for all samples (including thick ones) and parameters actually plot N/ρ (multiplied with ϕ_0 or not, that is less important)! This is very important information, and it is crucial to see if the peak in N/ρ actually coincides with the peak in N or not!

12) I 136 "violation of constant entropy with doping" - why would there be constant entropy with doping?

13) Fig. 3 a - the scales here are strange, is the bottom of the plots not 0? If not, then that offset should be visible!

14) I. 167 - the "uemura relation" is again just an upper bound (there are of course dopings and samples where T_c is much below the limit suggested by the superfluid density). Also one has to add that the data used by Uemura is somewhat cherry-picked and the relation is not quite as universal. So the observation of the authors is interesting in this context (i.e. possible superfluid-density bounds on T_c), but they should not describe the $T_c \propto 1/\lambda^2$ as a given and established thing. It is an upper limit scaling, and not always applicable. Let me know that I think this fact actually makes the authors' observations more interesting, not less interesting.

15) Fig 3d, how does the evaluated λ compare with other measurements e.g. based on $Hc1$?

16) Fig.4 - this may be a little much to ask for, but given that the authors have started putting data from other materials in their plot, and the full data is available - they should also check if the peak in N/ρ lines up with the peak in N and explain what value they use.

16b) What the authors CERTAINLY should do, is not just leave out the Nb-based superconductors from this plot! The Nb-based superconductors show a value of S that is MUCH larger than k_B , see this excerpt from Behnia JPCM: (" Huebener and Seher [56] studied high-purity foils and found a Nernst signal as large as $S_{xy} \sim \mu V K^{-1}$ in samples with a residual resistivity as low as $20 \text{ n}\Omega\text{cm}$. This would imply an entropy per vortex per sheet exceeding k_B by several orders of magnitude. In another study, de Lange and Otter [57] quantified the magnitude of the vortex transport entropy, S_d in a niobium alloy ($\text{Nb}_{80}\text{Mo}_{20}$) and found that it peaks to $S_d \approx 10-11 \text{ J K}^{-1}\text{m}^{-1}$. Combined with a lattice parameter of 0.3 nm , this yields a sheet entropy, $S_{\text{sheet}} d$ exceeding $200 k_B$ ") The authors cannot just leave this out, it would be very misleading! Whether they like it or not, the "universal" k_B is just not universal.

17) For the data from other materials, there should also be a proper table, with origin, field measured, N and ρ individually etc. Also some comment on

18) Fig. 4 Is the k -ET point in the right place? Behnia has it at almost exactly the same value,

In₂kB, as STO and FeSeTe. Do the authors use another calculation?

19) I 269 and SFig. 2 - this is the only serious technical problem I see: $T(R)$ of the temperature sensors is actually not single-valued and dR/dT is zero around about 20K - exactly a range where a lot of data was taken! How do the authors deal with this? Obviously a measurement-error in DT translates to a measurement error in N - so a serious analysis of systematic error propagation here is needed and the authors need to mention how they circumvent the non-single-valuedness and zero derivative!

20) Sfig 5 - as this is a 2D plot and the "onset temperature" is just a measure of signal to noise (see point 7 above)? Why do the authors not just plot something like $d(N/B)/dB$ or another quantity (e.g. the cubic component of a fit on $N(B)$) to show how a superconductivity-related Nernst effect approaches towards T_c ? This could actually be a rather interesting plot, of relevance for the main paper, and it is interesting indeed how T_c itself and the Nernst effect reduce non-proportionally.

21) Sfig2 - as the authors have actually collected Seebeck effect data - why don't they also show it? It would be useful for the community!

22) Sfig 3 - the bulk-like samples should also be analyzed fully for entropy, i.e. show N/ρ vs B and T , and also show the $\rho(T,B)$ data.

Some smaller points:

- I.25 "wide utilization" is not really the case for SC vortices.

- I47 and following, the axes (B along c etc.) need to be defined more clearly. Also an estimate of how well B and c are aligned (this is important as in-plane fields can have other results than out-of plane fields, see work by Logvenov for example).

- I 284 - I assume the authors mean "anti-symmetrize" B and $-B$ measurements but this should be made more clear.

- The authors should comment on the heat flow in sample vs substrate

In summary, I think this paper has great potential, but the presentation and high-level analysis does need to be improved.

Reviewer #1 (Remarks to the Author):

Hu *et al.* study the doping dependence of the transport entropy per vortex per unit length in a cuprate superconductor of BSCCO by measuring the magneto-resistivity and Nernst effect. To control the doping level while keeping the homogeneity of the sample, the authors utilized out-diffusion of oxygen from mechanically exfoliated ultrathin samples. Using the on-chip setup, the magneto-resistivity and Nernst effect were measured successfully for the ultrathin samples with different doping levels. It is found that with decreasing the doping level, the vortex transport entropy decreases monotonically from values close to $k_B \ln 2$ in the optimally doped regime to values two orders of magnitude smaller in the extremely underdoped regime, associated with the decrease in the superfluid density and transition temperature. The authors attribute the suppression of the vortex entropy to the emergence of the charge and spin ordering expected in the underdoped regime.

Measuring the thermoelectric effect in ultrathin samples is a challenging experiment and the reported data are probably reliable. The proposed scenario that the suppression of the vortex entropy originates from the emergence of charge and spin ordering in the underdoped regime looks interesting. However, I do not find clear experimental evidence to support their assertion. It is known that in the underdoped regime of cuprate superconductors, the charge ordering appears locally around quarter filling ($p = 0.125$), as reported in Ref. 33. However, the vortex entropy shown in Fig. 4 exhibits a monotonic decrease with decreasing the doping level. I would like to ask the authors if they observe the charge and spin ordering in their samples. Also, I would ask if they confirm that the vortex entropy takes values close to $k_B \ln 2$ in the overdoped regime, instead of showing a decrease. It is necessary for the authors to present data that show directly the correlation between the vortex entropy and the degree of order in the charge and spin ordered states, thus demonstrating that the suppression of the vortex entropy is indeed caused by the charge and spin ordering.

As mentioned above, the present manuscript lacks experimental evidence to support their claim and hence is not suitable for publication in Nature Communications. However, the observed suppression of vortex entropy with decreasing the superfluid density is interesting for the specialists. Therefore, it should be considered for publication in a specialized journal if the authors address the following points adequately.

[Our reply] We have substantially revised our manuscript to better convey the key findings of our work. Essentially, there are three discoveries:

1. The magneto-transport on ultrathin BSCCO demonstrates a clear dimensionality effect. In comparison, previous studies failed to find intrinsic properties in ultrathin BSCCO that are distinctly different from bulk samples. Our finding further allows us to extract quantitatively the superfluid phase stiffness/London penetration depth in the extremely underdoped regime. There was no data in this regime before.

2. We measure the Nernst effect over the complete underdoped side of the superconducting

dome. The Nernst signal shows nearly invariant peak value across this wide doping range but the extracted vortex entropy shows an exponential decay with decreasing doping. Such a systematic study provides important insight into the vortex dynamics and can draw broad interest from the superconductivity community and beyond. Our report is also the first one that challenges the recent proposal of a universal bound of vortex entropy, which has received much attention.

3. We analyze the superconducting fluctuations by obtaining the onset temperature and comparing our data with the theory of Gaussian superconducting fluctuations. Our experiment allows for a complete demarcation of the onset temperature on the underdoped part of the superconducting phase diagram. Such a systematic study was not achieved previously. Furthermore, we demonstrate that the theory of Gaussian fluctuations can be applied to understand the prominent Nernst effect above the superconducting transition temperature. We note that there existed no such comparative study in BSCCO before. These results can help settle down the debate on the role of superconducting fluctuations and the origin of pseudogap.

Although we relate the entropy reduction to the enhanced charge ordering, it is only one possible scenario in our discussions. The revised manuscript has made this point clear.

[1] In the sample S5, a peak position (T_{NP}) of the Nernst signal is well above $T_c = 5.4$ K. Then, the authors explained that the peak is caused by enhanced vortex fluctuations. This means that the vortices exist at temperatures up to 50 K. However, I have a serious concern about this interpretation. The authors mention in the introduction as: “As a result, the Nernst effect probes directly the vortex dynamics”. I consider that it is not correct exactly. The Nernst effect arises from three physical origins: (i) fluctuations of the phase in the superconducting order parameter, i.e., vortices, (ii) fluctuations of the amplitude in the superconducting order parameter, which cause the paraconductivity in the electric conductivity, and (iii) quasiparticles. The authors seem to ignore the contribution of (ii). Note that in Ref. 33, Cyr-Choinière *et al.* used the term “superconducting fluctuations”, instead of “vortex fluctuations”. I suggest the authors to discuss their Nernst signals on the basis of Ref. 19 in your manuscript.

[Our reply] Due to the controversy over the origin of Nernst signal above the superconducting transition temperature, we have corrected the labeling from “vortex fluctuations” to “superconducting fluctuations”.

Following the reviewer’s suggestion, we have added texts in the discussion of the main text about the superconducting fluctuations. These discussions are based on analysis added in Supplementary Note 7. In brief, our data find support to the scenario of Gaussian superconducting fluctuations.

[2] The authors obtained theoretical values of the vortex transport entropy using the equation (1), which is converted from a theoretical expression given by Sergeev *et al.* in Ref. 23. However, the equation (1) is probably incorrect since the authors ignore the temperature dependence of the coherent length. Thus, the resulting values of entropy in Fig. 4 should be reconsidered. Furthermore, in Ref. 24 Rischau *et al.* pointed out that the expression given by Sergeev *et al.*

leads to much larger values than experimental values ($\sim k_B \ln 2$) obtained by Rischau *et al.* The authors should explain why the equation (1) is valid.

[Our reply] Our purpose is to compare our experimental results with the theory of Sergeev, Reizer and Mitin. As explained in their work, the temperature dependence of ξ is neglected because they focused on the temperature regime of $T/T_c \in [0.2, 0.9]$. They pointed out that ξ in this temperature window “weakly depends on temperature”. We note that the temperature window is satisfied in our situation.

We have revised the descriptions related to the theoretical derivation of S_d to closely follow the derivation given by Sergeev, Reizer and Mitin. This revised derivation is now provided in the Method section. After this revision, the calculated vortex entropy at optimal doping is 2 to 4 times the experimentally extracted value, as shown in the revised Fig. 4a.

We note that Rischau *et al.* only calculated vortex entropy in Nb-SrTiO₃. BSCCO was not included there. The difference between theory and experiment may be material dependent.

[3] I cannot understand why the authors were able to measure the temperature difference between two thermometers below 20 K because the resistivity of the thermometer shows an upturn. The authors applied an ac current to a heater with a frequency of ω , producing 2ω oscillation in temperature with an amplitude of ΔT . ΔT is defined as deviation from the bath temperature T_{bath} of the cryostat. Seemingly, the authors used the value of T_{bath} to obtain the temperature difference. However, generally the minimum temperature during the temperature oscillation is larger than T_{bath} as shown in RSI 91, 095112 (2020) by Wu *et al.*

[Our reply] There are typically two types of thermometers: one with positive temperature coefficient (Pt-100, for example) and one with negative coefficient (Cernox from Lakeshore, for example). Our thermometer can be thought of as a combination of these two types, working at different temperature regimes. In the higher temperature regime ($T > 12$ K), our metal strip shows decreasing resistance with decreasing temperature, similar to typical Pt-100 thermometer. In the low temperature regime ($T < 12$ K), our metal strip shows increasing resistance with decreasing temperature, similar to the behaviors of RuO_x or indium oxide in RSI 91, 095112 (2020). We have added the following sentence to the Method section:

“We note that conventional on-chip thermometry loses sensitivity at sub-10 K regime due to the saturation of resistance. We have recently overcome this bottleneck and extended our thermometry down to 1 K by utilizing the Kondo effect⁵⁷.”

Ref. 57 above is our recent technical paper, in which we elaborated the working principle and the reason behind the up-turn: *Appl. Phys. Lett.* **120**,173507 (2022). We note that our thermometer loses sensitivity only around $T = 12$ K. This is why we do not have data points in the temperature window of [11, 13] K.

We thank the reviewer for pointing out that the minimum temperature at the two thermometers

during the heat oscillation is larger than the bath temperature. We have removed the misleading sentence in the Method section: “Consequently, the temperatures at the two thermometers (Th 1 and Th 2) deviated from the bath temperature of the cryostat by $\Delta T_{1,2}$ ”.

Apart from correcting the statements, we have also carefully checked the influence of such a deviation on our data analysis. Previously, we evaluated the temperature variation in the thermometers with a frequency of 2ω by using the equation: $\Delta T_{1,2}^{2\omega} = \Delta R_{1,2}^{2\omega} / \frac{dR_{1,2}}{dT}$, where the derivative was taken at the bath temperature: T_b . We have checked that using values of $\frac{dR_{1,2}}{dT}$ at the corrected temperatures for the two thermometers (Th1, Th2) (i.e., $T_{Th1/Th2} > T_b$) results in negligible change of the evaluated temperature gradient: δT_{1-2} .

[4] The authors mention that the vortex flow in the linear I-V regime is “flux flow”, where the pinning effect is absent. However, the observed Arrhenius law in the resistivity data clearly shows that the vortex flow is the thermally assisted flux flow (TAFF), where the pinning is effective [See Rev. Mod. Phys. 66, 1125 (1994)]. Thus, the authors should use the word TAFF. The discussion on the vortex entropy stays essentially unchanged whether the flow is flux flow or TAFF.

[Our reply] We thank the reviewer for this suggestion. We have revised our terminology accordingly.

[5] Line 183 on page 10, Fig. 3c -> Fig 3b

[Our reply] We have reorganized the figures and checked more carefully the labeling in the texts.

Reviewer #2 (Remarks to the Author):

The paper by Hu et al. reports a Nernst effect study of the vortex entropy in BSCCO superconductor. Here the authors show a decreasing entropy from optimal doping to underdoped regime in contrast to a universal value $k_B \ln 2$ as reported in ref.[24], and thus claim the breakdown of universal vortex entropy in underdoped BSCCO superconductor. However, it seems that the authors don't catch the main point about vortex entropy and are less rigorous to claim such a conclusion in this manuscript. Ref.[24] focuses on the study of Nernst effect below T_c , not above T_c , due to the different mechanism. Below T_c the vortex Nernst response dominates, but above T_c , the Nernst signal from the superconducting fluctuation strongly displays the material-dependent, and the amplitude of the Nernst signal over 6 orders of magnitude is not rare. Ref.[K. Behnia and H. Aubin, Rep. Prog. Phys. 79, 046502 (2016)]. In this paper the underdoped sample is $T_c \sim 5$ K, the comparable Nernst signal may be from the superconducting fluctuation not vortex. The obtained entropy in the normal state not superconducting state is not available to the equation in the paper. Thus it is not wonder that the underdoped sample shows different behavior. For these reasons, I do not recommend the publication of this paper in NC.

[Our reply] We would like to emphasize that we indeed used the data below T_c to calculate the vortex entropy in our original manuscript. We have further revised our manuscript to make this point clearer.

We highlight that it is the Nernst signal below T_c that yields an exponential decrease of vortex entropy with decreasing doping. This result cannot be explained by the superconducting fluctuations, which occur only above T_c .

Apart from the vortex entropy, we believe our paper has two other discoveries, which become clearer in the revised manuscript:

1. The magneto-transport on ultrathin BSCCO demonstrates a clear dimensionality effect. In comparison, previous studies failed to find intrinsic properties in ultrathin BSCCO that are distinctly different from bulk samples. Our finding further allows us to extract quantitatively the superfluid phase stiffness/London penetration depth in the extremely underdoped regime. There was no data in this regime before.

2. We analyze the superconducting fluctuations by obtaining the onset temperature and comparing our data with the theory of Gaussian superconducting fluctuations. Our experiment allows for a complete demarcation of the onset temperature on the underdoped part of the superconducting phase diagram. Such a systematic study was not achieved previously. Furthermore, we demonstrate that the theory of Gaussian fluctuations can be applied to understand the prominent Nernst effect above the superconducting transition temperature. We note that there existed no such comparative study in BSCCO before. These results can help settle down the debate on the role of superconducting fluctuations and the origin of pseudogap.

In addition, it is unbelievable that the vortex fluctuation (fig.5 in SI) in EUD sample can be

observed in such high temperature above T_c . Is still there a vortex liquid phase? Furthermore, the entropy density in the vortex may be different from the case of normal state.

[Our reply] By “vortex fluctuation”, we are actually referring to superconducting fluctuations. We have corrected this in our revised manuscript.

Noted that ρ_{xx} of Sample S4 in fig.2-a4 displays the largest value below T_c , almost twice larger than that of S5, but the peak of Nernst signal of S3, S4, and S5 is almost same in fig.2 c3-c5. Why the S_d (sheet) of sample a4 in fig 3b does not be the minimum?

[Our reply] We thank the reviewer for carefully checking our manuscript. The normal-state resistivity at $T > T_c$ is indeed larger in S4 than that of S5. However, we highlight that the vortex entropy is evaluated by using the data below T_c .

Reviewer #3 (Remarks to the Author):

In their manuscript, Hu and coauthors report on measurements of the Nernst effect and of the magnetoresistance in ultrathin flakes (and thicker samples) of BSCCO in a range of dopings, from optimally doped (OD) to strongly underdoped (EUD). They focus on using this data to extract the vortex entropy by taking a ratio of Nernst effect and the resistivity and find a very strong reduction of this ratio, by several orders of magnitude, when going from OD to EUD. They relate this to recent results by Behnia's group (Rischau et al. PRL2021) who had suggested a univesal bound (and in that paper, an **_upper_ bound** was postulated) on the entropy per vortex per layer of $\ln(2)k_B$.

Upon first reading the abstract, this paper seemed to me like a clear case for rejection - after all the upper bound postulated by Richau et al. was supposed to be an upper bound (and was problematic in itself, see below), so seeing values below that should not come as a surprise, could have many reasons (e.g. residual non-flux-flow resistivity), and does appear in materials that hit the "upper bound" when looking at other points in the B-T space. I still believe that in terms of the "high-level analysis" the presentation in this paper cannot remain as it is, and I could certainly not recommend publication in its current form. However, looking at the quality of the data itself, how it was taken, and the "low-level analysis", this paper is probably the best and most careful measurement of the Nernst effect in a superconductor I have seen so far.

So if the authors are willing to make the effort to put their work into a proper context, and remove the misleading claims and incorrect generalizations that currently plague the work, I believe the paper could be an extremely valuable contribution to the study of the Nernst effect in superconductors - a field of study which is both still very badly understood (so experiments are important), and very controversial (especially above T_c , and this indeed was caused by the initial rather careless presentation of results by Ong's group). It is therefore very crucial that the authors are very honest and precise in their statements. In such a case however, I could certainly see how this could become a work that should be published in this journal. In the following I will try to provide the information and comments necessary so that the authors can make the necessary improvements to bring the work to this level.

[Our reply] We would like to thank the reviewer for the careful evaluation of our work and for providing very helpful remarks and comments. We have greatly benefited from learning this report, which helps elucidate many controversial and ambiguous points in this field. We have closely followed the reviewer's comments and substantially revised our manuscript.

Let me provide a bit of context: In Rischau et al. PRL2021 the authors made two claims. The first is that, despite otherwise extremely different parameters (e.g. gap and T_c , lattice constant etc.), the peak (in B-T space) Nernst signal N in numerous superconductors lies in a range somewhere between 1 and 10 $\mu\text{V}/\text{K}$. Although this claim does not entirely hold up, - there are some reports in organic κ -(BEDT-TTF) $_2$ X $_2$ samples which rather range in the tens of nV/K (<https://www.nature.com/articles/nature06182> and <https://www.nature.com/articles/srep03390>), although one may question those given other

measurements in the same material that report higher values ([https://doi.org/10.1016/0921-4534\(96\)00243-2](https://doi.org/10.1016/0921-4534(96)00243-2)) - it does still seem to be strikingly general. In a later work by Behnia (J. Phys.: Condens. Matter 35 (2023) 074003, also cited - I will refer to it as Behnia JPCM) this value is related to a viscosity-to-entropy density ratio, and this is certainly a topic that deserves further exploration.

The second claim in that work is that when dividing the Nernst signal at its peak by the resistivity at the same field and temperature, ρ , a universal bound of $\ln(2)k_B$ per vortex per layer (in that case the "layer" is taken as one lattice constant in the field direction) is not exceeded (and the values reported in that work seem to hit that limit to two significant digits).

The latter claim had two problems. First of all, if the ratio of N/ρ gives access to the entropy S , why should one consider its value at the peak of N ? Instead, one should consider the peak of N/ρ itself. As one can see for example in work by Li et al (Chin. Phys. B Vol. 29, No. 8 (2020) 087402), also cited by the authors, this ratio can actually be much larger at lower fields than it is at the peak of N . If one wants to make general statements about vortex entropy, there is no reason to limit those studies to the peak of N , which is ultimately an experimental quantity (whereas the ratio of N/ρ gives α_{xy} , which is more intrinsic). Second, there are actually a number of materials where this "universal bound" of $\ln(2)k_B$ is exceeded, and in Behnia JPCM2023 those works are indeed mentioned. The value in most of these is still "around k_B " (and there is some uncertainty of what constitutes the length-scale of one layer).

However, importantly, in Niobium-based superconductors, the entropy exceeds $200k_B$! (see section "Historical Nernst data on superconducting niobium and its alloys" and refs 56 and 57 in that paper. So one can clearly say that the "universal bound" for S postulated by Rischau et al. simply didn't hold up to further scrutiny, and the current manuscript submitted here is a further confirmation that $\ln(2)k_B$ is not a valid upper bound. As for a lower bound, that was not postulated there (although in Behnia JPCM there is a brief speculation, but it is enough to take any data at e.g. higher B-field, see for example Li et al, or other temperatures, to see that this does not hold up experimentally - although it is of course not clear in those other works if some non-flux-flow resistivity also comes in).

[Our reply] We agree with the reviewer that $k_B \ln 2$ is not a valid upper bound. Nevertheless, the scenario of a lower bound, as claimed in Behnia JPCM, seems to be valid in the experimental systems that have been explored by so far (before our work).

The reviewer argued against the lower bound scenario by pointing out N/ρ further decreases at higher fields. However, as the reviewer also recognized, this further decrease, after N reaches the peak, can have a mundane origin. Simply, superconductivity in this regime is already strongly suppressed and the flux-flow situation is not applicable. In Fig. R1, we show the data from our own experiments on NbSe₂. N reaches a peak when $\rho(\rho_{xx})$ is already 80% of the normal state resistance. The sample may turn partially non-superconducting. The drop of Nernst signal at higher magnetic field reflects suppressed vortex contribution.

Fig. R1 Nernst signal N of a NbSe_2 flake as a function of magnetic field B at 5 K. As a comparison, the resistivity ρ_{xx} is also displayed.

From the work of Li *et al.* (ref. 21 in the updated manuscript) N/ρ keeps decreasing with increasing magnetic field and smoothly crosses the position where N itself has a peak (Fig. 2 of ref. 21). Taking the value of N/ρ at the peak position of N may therefore mark a lower bound.

In our revised manuscript, we have followed the suggestion of the reviewer by showing N/ρ over the entire temperature regime. With decreasing doping, there still exists a general decreasing trend of N/ρ below T_c .

So certainly the authors cannot say, as they do in title/abstract, that theirs is an observation of a breakdown of some universal vortex entropy value. As mentioned above, the only thing that does seem somewhat universal is the peak value of N itself, and there the values that the authors find are between 0.8 $\mu\text{V}/\text{K}$ and 2 $\mu\text{V}/\text{K}$, so pretty much within the expected range (again, surprising given the very large change in T_c , so an interesting result in itself).

[Our reply] Although we still believe that our data in the flux flow regime suggests a clear breakdown of the lower bound proposed by the Behnia group, we realize that the very existence of a lower bound seems still controversial and the Nernst effect from our study may find broader interest. We thus follow the suggestion of the reviewer and have changed the title of our manuscript.

We thank the reviewer for pointing out the importance of a nearly invariant Nernst signal itself. We have added sentences in the main text to highlight this observation.

Nevertheless I believe that the authors' data is very valuable, and I will briefly mention what I find is very carefully done, in contrast to many other measurements, including recent measurements, of the Nernst effect in superconductors: There is a rho curve associated with every N curve, the parameter range is exhaustive, linearity is checked, it is convincingly argued how and where rho is of flux-flow type, the thermal gradients used are very small, the geometry is carefully chosen, and importantly (but unfortunately not the case in many papers), all the details of the measurement are given.

[Our reply] We thank the reviewer for appreciating the importance of our work. We have

significantly revised our manuscript according to the points raised. Below, we provide a point-to-point reply.

However, there are a number of important points that the authors need to address (roughly in order of appearance in the paper, all are important):

1) As mentioned the title/abstract cannot stand like this. What they show is a careful and exhaustive study of the Nernst effect, Magnetoresistance and their ratio in a wide range of parameters (doping AND thin vs thick) in an underdoped cuprate. It is an important contribution to the study of vortex entropy, but it is NOT a breakdown of a universal value.

[Our reply] We have provided our reasonings above for the possible lower bound of vortex entropy. We therefore still believe that our results constitute a serious challenge to this claim and will attract broad interest. However, we have changed the title and revised the abstract to focus not only on the vortex entropy. In the revised manuscript, we also emphasize on our discovery of dimensionality effect in ultrathin BSCCO and the analysis of superconducting fluctuations.

2) I. 49 What the authors ignore here is that in addition to the thermal force term there is also a Magnus force term (see Krasnov and Logvenov *Physica C* 274 (1997) 286-294 for theory, and [https://doi.org/10.1016/0921-4534\(96\)00243-2](https://doi.org/10.1016/0921-4534(96)00243-2) for experiments). The authors need to consider that force as well, and either take it into account or argue why it is small and can be ignored

[Our reply] We have revised the introduction part and added the discussion on the Magnus force to the method section:

“In deriving the formula for transport entropy per vortex $S_d = N\Phi_0/\rho_{xx}$, it is often assumed^{18,19,21} that the thermal force on a vortex is balanced by viscosity such that $-S_d \nabla_x T = \eta v_x$, where η is the viscosity coefficient. The Nernst signal is therefore:

$$N = BS_d/\eta. \quad (1)$$

However, the drifting vortex can experience the Magnus force. Consequently, it is the total force—the sum of thermal force and Magnus force—that becomes balanced by the viscous force. H.-C. Ri *et al.* considered both the above-mentioned effect and the Hall effect of unbound quasi-particles⁵⁹. They obtained the following equation for the Nernst signal:

$$N = \frac{S_d \rho_{xx}}{\Phi_0} + S_n \frac{\rho_{xx}}{\rho_n} (\tan \theta_{QP} - \tan \theta_V), \quad (2)$$

where S_n is the Seebeck coefficient in the normal state, ρ_n is the normal state resistivity, θ_{QP} and θ_V are the Hall angles for vortices and quasi-particles, respectively. Equation (2) indicates that there exists an additional term that may contribute to the Nernst signal. We note that the two Hall angles are equal in the Bardeen-Stephen model⁵⁹ such that the second term is zero. Furthermore, the second term has a different temperature dependence than the first one in Eq. (2). If this term has a noticeable contribution, the temperature dependent Nernst effect should exhibit a shoulder instead of a single peak⁵⁹. Experimentally, we observe no such a shoulder in our data (Fig. 3b1-b5). We therefore neglect the second term in evaluating the vortex entropy.”

3) Fig.1 , it is not clear how ρ_{13} or ρ_{24} are extracted, given that they are transverse to the bias current. In general there should be more details on the 4-point measurement beyond what is given (on how geometry is factored in). Also, it is not clear exactly what the measured zero value is in this plot. Is any kind of background subtracted? If so, that should be stated clearly.

[Our reply] We note that no background was employed. We have added dashed horizontal lines in the figure to mark the zero value for each curve. The reason why we added a small offset for different traces of the same sample is because otherwise the curves are nicely overlapping.

As regard to ρ_{13} and ρ_{24} , we have changed to use resistance instead of resistivity for marking all the traces. Here, R_{13} and R_{24} mainly stems from the longitudinal resistance that mixed into the transverse measurement due to the non-ideal geometry of the flake. This explanation has been now added to the figure caption.

In our experiment, the longitudinal resistivity is calculated as: $\rho = \frac{V_{AB} t w}{I l}$, where V_{AB} is the measured voltages between the A and B contacts, I is the input current, w and t are the sample width and thickness, l is the center-to-center separation between the voltage contacts along the longitudinal direction. This simple formula assumes uniform flow of current in a regularly shaped sample. It should be valid for our thick flakes, because they have nearly rectangular shapes as shown in Fig. R2. However, our ultrathin samples have clear trapezoid shapes between the source and drain contacts (Fig. R3a). In this case, the resistivity should be evaluated as: $\rho = \frac{V_{AB} t}{I g}$, where g is a geometric factor and its value may deviate from the simple length-to-width ratio l/w .

Fig. R2 Optical image of a bulk-like BSCCO sample (S7). Scale bar: 20 μm .

To evaluate g , we carry out finite element analysis. An example is given in Fig. R3b. The shape of the sample and the arrangement of gold contacts (marked by black lines) are based on the optical image of our sample. The color-coded plot illustrates the voltage distribution in this sample. Table R1 shows that l/w and g have similar values for each sample. Therefore, simply taking the l/w ratio does not affect the evaluation of resistivity. We note that our finite element analysis neglects possible contact resistance and assumes straight boundaries of the flakes. In order to avoid further complication, we use the simple l/w ratio for evaluating the resistivity in our

experimental analysis.

Fig. R3 a, Schematic drawing of a typical ultrathin BSCCO flake. **b**, Electric potential distribution across the area as illustrated by the red frame in panel **a**. We choose the length between voltage contacts A and B to be the unit length. The resistivity of sample is set to be 1 and the resistivity of the gold electrodes (demarcated by black lines) is set to be 0.01.

Table. R1 Geometric factors of ultrathin BSCCO samples studied in this work.

Sample No.	l/w	g
S1	0.228	0.233
S2	0.147	0.112
S3	0.180	0.178
S4	0.257	0.245
S5	0.287	0.274

4) How is T_c really defined? What does "reaches zero" mean? Reaches the signal-to-noise level? The authors should be very explicit here and also make it clear if there is any change depending on the method chosen to determine T_c (is a threshold or ρ_{100K} used? or an extrapolation of the curve?) Also, many authors rather chose the downturn or midpoint of the transition as their T_c (see also below), the author should comment on their choice.

[Our reply] In the revised manuscript, we employ a more quantitative method to define T_c . Namely, T_c is defined as the temperature where the resistivity reaches 1% of the normal state resistivity (ρ_N). We refrain from using the midpoint or onset. This is consistent with other studies on cuprates, such as ref. 30 in the updated manuscript.

We have provided details on the definition and comparison to other criteria in the added supplementary note 5. For convenience, we show the related figure as Fig. R4 below. The definition of T_c by using the 1% standard finds further support from the comparison of $T_c(B)$ curve and the Nernst signal, as we illustrate in the answer to point 5.

Fig. R4 a-e, Linear extrapolation of the resistivity curve of S1-S5 showing the extraction of the normal state resistivity. The intersection point of the resistivity curve and the 1%, 50%, 90% ρ_N line gives the value of $T_c^{1\%}$, $T_c^{50\%}$ and $T_c^{90\%}$, respectively.

5) The authors should plot (SI is fine) $T_c(B)$, comment on how that compares to other measurements in BSCCO and if the field-dependence of N maps to the field-dependence of $T_c(B)$ (i.e. when plotting N/B in B - T space, do the contour lines of N/B follow $T_c(B)$) Such contour plots, both for N itself and for N/B , should be included also.

[Our reply] Part of the answer to this point has been given in the answer to point 4 above. We have added the requested plot of $T_c(B)$ and N in T - B space in the supplementary information (see Supplementary Fig. 10). The field-dependence of $T_c^{1\%}(B)$ matches well the contour of the Nernst signal at the lower temperature side of the Nernst peak. It suggests that $T_c^{1\%}(B)$ can reflect the vortex-melting curve. By contrast, $T_c^{50\%}(B)$ or $T_c^{90\%}(B)$ do not show direct correspondence with the contour of N . This is possibly due to the mixed contribution to ρ from both vortex motion and quasi-particles, whereas the vortex motion dominates the contribution to N at $T < T_c^{1\%}(B = 0)$. Similar conclusions can be reached by comparing the Nernst coefficient $\nu \equiv N/B$ with $T_c(B)$.

6) The authors should comment on the large width of the transition even at $B=0$. How does it compare to the bulk-like samples (the $\rho(B,T)$ curves should also be shown there!)? Do they think the width is related to the low dimensionality, or rather to inhomogeneity? Does BKT physics come in? That should be mentioned.

[Our reply] We compare the resistive transition of ultrathin samples with that of bulk-like samples in Fig. R5. Near optimal doping (OP), we find that the ultrathin sample has a broader transition than the bulk-like ones (Fig. R5a). In the underdoped (UD) case, however, our ultrathin samples possess a comparable or even slightly narrower transition than the bulk-like samples (Fig. R5b). This latter improvement, as we pointed out in the revised manuscript (Fig. 2), mainly stems improved sample quality. Namely, the as-grown underdoped crystal has stronger inhomogeneity than the underdoped ultrathin sample realized by natural release of oxygen. From this perspective, it is tempting to attribute the broadened transition near OP to enhanced inhomogeneity as well. However, as shown in Fig. 2a,f, sample S1 possesses similar normal state resistivity to that of sample S6. It suggests that the carrier mobility in the ultrathin sample is comparable to that of thick ones. We therefore speculate that the broadening of the resistive transition near OP may stem from enhanced thermal fluctuation effect in 2D, similar to the case of NbSe_2 [*Nat. Phys.* **12**,

We have also investigated the possible involvement of BKT transition. As shown in Fig. R5c-h, the resistive transitions in both ultrathin and bulk-like samples can be fit with the Halperin-Nelson equation. It suggests that the BKT transition already occurs in bulk-like samples due to strong anisotropy of BSCCO. This is consistent with previous studies that confirmed the BKT transition even in bulk crystals of BSCCO [*Phys. Rev. B* **47**, 11374 (1993)]. In fact, no previous experiments identified any intrinsic properties in ultrathin BSCCO that are different from bulk crystals [*Nature* **575**, 156-163 (2019), see also ref. 34 in the updated manuscript]. Our finding of 2D vortex creeping in Fig. 2h-j therefore constitutes first evidence for the long sought-after dimensionality effect.

Fig. R5 a, Normalized resistance as a function of temperature for S1 and S6. **b**, Width of the superconducting transition $\Delta T \equiv T_c^{90\% \rho_N} - T_c^{1\% \rho_N}$ for sample S1-S8. **c-h**, $(d\ln R/dT)^{-2/3}$ as a function of T for S1-S6. Red lines are linear fits. Intercepts of them with abscissae yield the BKT transition temperature: T_{BKT} . Insets in **c-h** compare experimental curves with the BKT fitting: $R = R_0 \exp(-b(T - T_{BKT})^{-1/2})$, where R_0 , b are fitting parameters. Red dots mark the position for $T_c^{90\% \rho_N}$.

7) I. 125 and others: Cyr-Chronière et al. do NOT claim that the onset of a B-dependent N/B signals the onset of `_vortex_` fluctuations. It is important to be precise and clear here, all statements related to that need to be changed. In the cited PRB, Cyr-Chronière et al. actually remain agnostic to whether the B-dependent N/B they see can be explained by Gaussian fluctuations or not, all they claim is that the B-dependence of this signal (i.e. a non-linear dependence of N on B) is the signal of a Nernst effect that is related to superconducting fluctuations, rather than being purely quasiparticle-driven. Furthermore, the "onset" of something is of course just a measure of the signal to noise of the experiment! So the authors need to be clear on how they actually define the point they choose - or they better decide not to choose one specific temperature, but just to show how $d(N/B)/dB$ actually changes vs temperature!

[Our reply] We have changed the statements from "vortex fluctuations" to "superconducting fluctuations". We have also adopted a quantitative definition for the onset temperature of superconducting fluctuations. In short, we define the onset as the temperature where the nonlinearity of $N(B)$ exceeds the experimental noise level. We provide details in Supplementary Note 6.

In addition, we have employed the temperature where the slope of $N(B)$ changes sign as another indicator for the dominance of quasi-particles, because the contribution from superconducting fluctuations cannot change sign. As shown in Fig. 4b of the revised manuscript, both definitions support the similar conclusion that superconducting fluctuations persist only in a limited range, which is far below the pseudogap temperature.

8) In this context, the authors should make the effort and compare their results above T_c (and here they should consider different possible definitions of T_c , including onset or midpoint) to the theory of Gaussian fluctuations. This is still an open and hot debate, and as the data presented here is of high quality, it would be important to consider this data in this context. The authors should be aware that the original work on the Nernst effect resulting from Gaussian fluctuations has by now been extended quite a bit (to also include nonlinearities), see Glatz et al. Phys. Rev. B 102, 174507 (<https://journals.aps.org/prb/abstract/10.1103/PhysRevB.102.174507>) for a recent work that also summarizes previous works. This is an important point.

[Our reply] We have provided our reasons for the choice of T_c in the answers to points 4 and 5. Following the request of the reviewer, we have analyzed our data, especially for sample S3 to S5 with sufficient data points in the superconducting fluctuation regime, in the framework of Gaussian fluctuations. Our main finding is that the theory of Gaussian fluctuations can well describe the Nernst signal above T_c in a certain temperature window. We have added a paragraph in the discussion section of the main text, together with supplementary note 7, on this important point. We believe this addition has substantially strengthened the paper and we thank the reviewer for the thoughtful suggestion.

In the answer to Point 5, we have provided reasons for defining T_c as the point where resistance drops to 1% normal state. We show in Fig. R6 below the comparison of Gaussian fluctuations

Fig. R6 a,b Nernst coefficient in the zero-field limit, v_0 , divided by the zero-field resistivity ρ_0 for sample S5 as a function of the reduced temperature $t (= T/T_c - 1)$. Here T_c is defined as the point where the resistance drops to 1% (panel **a**) or 50% (panel **b**) of the normal state resistance. Dashed lines are obtained by inserting the extracted coherence length ξ_{ab}^0 (6 and 3 nm for **a** and **b**) into the equation for Gaussian superconducting fluctuations (Eq. S2 in the supplementary information).

analysis by using 1% and 50% ρ_n as the definition of T_c . We carry out this comparison for sample S5. There is a lack of data set in the superconducting fluctuation regime for S3 and S4 if choosing 50% ρ_n for defining T_c . In general, changing the definition does not significantly affect the conclusion: the general trend predicted by Gaussian fluctuations (dashed lines) can still be seen by using the midpoint for T_c . Quantitatively, the fitting parameter—the superconducting coherence length—may vary.

9) I.131 How does being in a linear regime imply that vortex pinning is negligible (cf also Supplementar Note 3)? Sure, if all pinning centers had exactly the same pinning force I would expect a sharp threshold, but given that pinning centers are probably widely distributed in their pinning force, that is not the case. A more careful discussion is needed here. It is almost impossible that there is no pinning in the underdoped samples, given the intrinsic disorder that has to exist there. It is good however that the authors have explicitly tested for linearity here!

[Our reply] We have revised the related discussions. Essentially, we want to emphasize that both the Nernst signal and the magneto-resistivity are measured in the linear response regime. Within the experimental range for temperature difference and ac current, we do not see non-linear behaviors.

10) In this context, how does $\rho(B)$ compare with $\rho(\text{normal})$ when plotting it as $\rho(B/B_{c2})$ - if the authors have access to B_{c2} ($H_{c2}...$) either from the extrapolation mentioned above or from literature if available?

[Our reply] The requested plot is shown in Supplementary Fig. 8. We compare $\rho_{xx}(B)$ with the flux-flow resistivity $\rho_f(B) = \rho_N B/B_{c2}^0$ based on the Bardeen-Stephen model at the same temperature for each sample. The trend of $\rho_{xx}(B)$ agrees well with the flux-flow behavior.

11) I.134 "typically evaluated" is not really the case, it is still only a few works. In any case and this is a VERY important point: The authors should, for all samples (including thick ones) and parameters actually plot N/ρ (multiplied with ϕ_0 or not, that is less important)! This is very

important information, and it is crucial to see if the peak in N/ρ actually coincides with the peak in N or not!

[Our reply] We have substantially revised the manuscript and updated the corresponding figures with the analysis of N/ρ . The peaks in N/ρ and N occur at different temperatures: $T_{N/\rho}^p < T_N^p$. Nevertheless, the values of vortex entropy calculated at either $T_{N/\rho}^p$ or T_N^p (if both temperatures are below T_c) are on the same order of magnitude.

12) I 136 "violation of constant entropy with doping" - why would there be constant entropy with doping?

[Our reply] We have revised the corresponding texts. "A constant entropy" mainly refers to the lower bound indicated by several distinct superconductors. Theoretically, a nearly invariant vortex entropy with doping is expected because: $S_d^{cal} \sim \frac{\Phi_0^2}{4\pi\mu_0\lambda^2 T_c}$, and we have confirmed experimentally that $1/\lambda^2 \propto T_c$.

13) Fig. 3 a - the scales here are strange, is the bottom of the plots not 0? If not, then that offset should be visible!

[Our reply] The plot mentioned here has been changed into Supplementary Fig. 8, which have been revised to clarify the scales.

14) I. 167 - the "uemura relation" is again just an upper bound (there are of course dopings and samples where T_c is much below the limit suggested by the superfluid density). Also one has to add that the data used by Uemura is somewhat cherry-picked and the relation is not quite as universal. So the observation of the authors is interesting in this context (i.e. possible superfluid-density bounds on T_c), but they should not describe the $T_c \propto 1/\lambda^2$ as a given and established thing. It is an upper limit scaling, and not always applicable. Let me know that I think this fact actually makes the authors' observations more interesting, not less interesting.

[Our reply] We have followed this suggestion and revised the corresponding descriptions in our manuscript.

15) Fig 3d, how does the evaluated λ compare with other measurements e.g. based on H_{c1} ?

[Our reply] We have compared the evaluated λ with that from other measurements (The corresponding table is provided in the supplementary information). This has resulted in the updated Fig. 2j. We emphasize that there is currently a lack of studies on BSCCO in the extremely underdoped case. Our extracted λ may therefore guide future investigations.

16) Fig.4 - this may be a little much to ask for, but given that the authors have started putting data from other materials in their plot, and the full data is available - they should also check if

the peak in N/ρ lines up with the peak in N and explain what value they use.

[Our reply] We plot the resistivity ρ , the Nernst signal N and $\alpha_{xy} = N/\rho$ in Supplementary Figs. 13, 14. From the plots, different superconducting materials can be roughly divided into three types on the basis of the trends of N and α_{xy} . The first type includes κ -ET and LSCO, where a peak feature occurs in both N and α_{xy} and the peak positions are very close. The second type includes YBCO and BSCCO films, where the peak in α_{xy} is usually at a lower temperature than that of N . Our data on BSCCO is consistent with this scenario. In the last scenario, including STO, FeSeTe, NCCO and NbSe₂ (see ref. 21 in the updated manuscript), the peak feature is only present in N but absent in α_{xy} . The existence of three different types also seems to challenge the claim of a universal bound of vortex entropy, consistent with the message we would like convey based on our data from BSCCO. We have provided a table (Table S2 in the supplementary information) that lists the values we use for plotting Fig. 4a.

16b) What the authors CERTAINLY should do, is not just leave out the Nb-based superconductors from this plot! The Nb-based superconductors show a value of S that is MUCH larger than k_B , see this excerpt from Behnia JPCM: (" Huebener and Seher [56] studied high-purity foils and found a Nernst signal as large as $S_{xy} \sim \mu\text{V K}^{-1}$ in samples with a residual resistivity as low as 20 n Ωcm . This would imply an entropy per vortex per sheet exceeding k_B by several orders of magnitude. In another study, de Lange and Otter [57] quantified the magnitude of the vortex transport entropy, S_d in a niobium alloy (Nb80Mo20) and found that it peaks to $S_d \approx 10\text{--}11\text{ J K}^{-1}\text{m}^{-1}$. Combined with a lattice parameter of 0.3 nm, this yields a sheet entropy, S_{sheet} exceeding 200 k_B) The authors cannot just leave this out, it would be very misleading! Whether they like it or not, the "universal" k_B is just not universal.

[Our reply] The data point for Nb₈₀Mo₂₀ has been added to Fig. 4a in the revised manuscript. As the plot indicates, the data from the previous studies seem to be all close or above $k_B \ln 2$. In this regard, our experiment seems to be the first one that shows entropy values that are far below $k_B \ln 2$.

17) For the data from other materials, there should also be a proper table, with origin, field measured, N and ρ individually etc. Also some comment on

[Our reply] We have added these values in Table S1 of the revised supplementary information.

18) Fig. 4 Is the κ -ET point in the right place? Behnia has it at almost exactly the same value, $\ln 2 k_B$, as STO and FeSeTe. Do the authors use another calculation?

[Our reply] We have taken the value of κ -ET ($S_d^{\text{sheet}} = 0.96 \times 10^{-23} \text{ J/K}$) from the study done by K. Behnia [JPCM 35, 074003 (2023)]. As shown in the revised Fig. 4a, it is clear that the quantity is close to $k_B \ln 2$.

19) I 269 and SFig. 2 - this is the only serious technical problem I see: $T(R)$ of the temperature sensors is actually not single-valued and dR/dT is zero around about 20K - exactly a range where

a lot of data was taken! How do the authors deal with this? Obviously a measurement-error in DT translates to a measurement error in N - so a serious analysis of systematic error propagation here is needed and the authors need to mention how they circumvent the non-single-valuedness and zero derivative!

[Our reply] As we explained in a technical paper before [*Appl. Phys. Lett.* **120**, 173507 (2022)], the resistance (R) of our thermometers exhibits an upturn at temperatures below 12 K. Although the resistance is not single-valued, we carry out thermoelectric measurements in the two temperature windows: [1.6, 10] K and [15, 300] K separately. Around 12 K (about 11-13 K), the thermometers lose temperature sensitivity. This is the temperature window that we do not have data points.

In our experiments, the noise level of voltages recorded by lock-in amplifiers is around 10 nV. This voltage noise induces an error of $\delta(\Delta R) = 0.1 \text{ m}\Omega$ in the measured resistance variations $\Delta R_{1,2} = \Delta V_{1,2}/I_{1,2}$ of the thermometers. From this evaluation, the error in ΔT ($\delta(\Delta T)$) can be readily obtained from $\delta(\Delta T) = \delta(\Delta R) / |dR_{1,2}(T_b)/dT_b|$. At $T_b = 10 \text{ K}$, for example, $|dR_{1,2}(T_b)/dT_b|$ is 22.2 m Ω /K such that $\delta(\Delta T) = 4.5 \text{ mK}$. This value is much smaller than $\Delta T = 80 \text{ mK}$ that we employed at this temperature.

20) Sfig 5 - as this is a 2D plot and the "onset temperature" is just a measure of signal to noise (see point 7 above)? Why do the authors not just plot something like d(N/B)/dB or another quantity (e.g. the cubic component of a fit on N(B)) to show how a superconductivity-related Nernst effect approaches towards Tc? This could actually be a rather interesting plot, of relevance for the main paper, and it is interesting indeed how Tc itself and the Nernst effect reduce non-proportionally.

[Our reply] We have employed a more quantitative protocol to define the onset temperature for superconducting fluctuations. This is now explained in the added supplementary note 6.

21) Sfig2 - as the authors have actually collected Seebeck effect data - why don't they also show it? It would be useful for the community!

[Our reply] Supplementary Figure 10 displays the Seebeck coefficient $S(T)$ from ultrathin ($\leq 2 \text{ UC}$) BSCCO flakes S1-S5. The general temperature dependence, i.e., a broad peak at around 100 K (or 150 K) with a linear decreasing trend at higher temperatures, and the exponential decay of normal-state S with p are consistent with previous studies on bulk BSCCO samples [S.D. Obertelli *et al.* PRB **46**, 14928 (R) (1992), F. Munakata *et al.* PRB **45**, 10604 (1992)]. Moreover, the value of S in the ultrathin samples is comparable to that in the bulk counterparts with the same T_c .

22) Sfig 3 - the bulk-like samples should also be analyzed fully for entropy, i.e. show N/rho vs B and T, and also show the rho(T,B) data.

[Our reply] We have added the analysis of N/ρ for the bulk-like samples in Supplementary Fig. 5. The resistivity data of bulk-like samples are shown in Fig. 2 of the main text and Supplementary Fig. 3. We have now pointed out in the main text that the magneto-resistivity of bulk-like samples is

complicated by current redistribution at different magnetic fields. Consequently, the measured ρ_{xx} in bulk samples may not originate from in-plane transport alone. In the revised manuscript, we focus on analyzing the vortex entropy in ultrathin samples only.

Some smaller points:

- I.25 "wide utilization" is not really the case for SC vortices.

[Our reply] This description has been removed in the revised manuscript.

- I47 and following, the axes (B along c etc.) need to be defined more clearly. Also an estimate of how well B and c are aligned (this is important as in-plane fields can have other results than out-of plane fields, see work by Logvenov for example).

[Our reply] We carried out measurements by using a rotatable sample holder from Oxford Instruments. It can rotate from -130 degrees to 130 degrees, with 0 degree marking the position where the sample is supposed to be perpendicular to the magnetic field. In our experiments, we always set the rotator to the angle of zero degree.

In order to check the field alignment, we studied superconducting NbSe₂ flakes by using the same sample holder. One representative result is summarized in Fig. R7 below. At a temperature right at the superconducting transition (6.6 K, marked as dashed line in Fig. R7a), applying 2 T at 0 degree can bring NbSe₂ fully to the normal state. However, rotating to the perfect parallel field direction gives rise to a sharp dip (Fig. R7b). This is because NbSe₂ is a strongly anisotropic superconductor such that the in-plane upper critical field is much larger than the out-of-plane field. From this angular dependent study, we can obtain that the dip reaches its minimum at $\theta = 91^\circ$. Similar angular dependent studies have been carried out on multiple samples and the resistance minimum emerges at $\theta = (90 \pm 1)^\circ$. It indicates that the misalignment between the nominal angle from the instrument and the real angle of the sample in the magnetic field is within 1°.

The contribution from a small in-plane magnetic field should be negligible in our experiment. Even if there exists Josephson vortices introduced by the in-plane magnetic field, their motion

Figure R7 a, Longitudinal resistance R_{xx} as a function of temperature under perpendicular magnetic fields $B = 0$ T (blue curve) and 2 T (red curve), respectively. Inset shows an optical image of the NbSe₂ sample. Gray vertical line marks $T = 6.6$ K. **b**, R_{xx} as a function of the nominal rotation angle θ at $T = 6.6$ K and $B = 2$ T.

under an in-plane temperature gradient would only give rise to Nernst signal in the out-of-plane direction, which does not affect our measurement by using bottom contacts.

- I 284 - I assume the authors mean "anti-symmetrize" B and -B measurements but this should be made more clear.

[Our reply] We have added the following description at the corresponding place: "This is realized via: $N(B) = [N_{raw}^+(B) - N_{raw}^-(-B)]/2$, where $N_{raw}^+(B)$ and $N_{raw}^-(B)$ are traces taken at positive and negative magnetic fields, respectively."

- The authors should comment on the heat flow in sample vs substrate

[Our reply] In our work, heat is generated by the on-chip heater and then propagates through the SiO₂ layer (300 μm thick) and the BSCCO samples, both of which possess low thermal conductivity (SiO₂: ~ 0.5 W/m • K [*Microscale Thermophysical Engineering* **8**, 1-5 (2004)]; BSCCO: ~ 2 W/m • K [*Supercond. Sci. Technol.* **33**, 025006 (2020)]). It gives rise to a temperature gradient across the samples. Finally, the heat flow can also dissipate in the Si substrate and the h-BN layers on top of BSCCO flakes. Si and h-BN both have a higher thermal conductivity (Si: 150 W/m • K [*Microscale Thermophysical Engineering* **8**, 1-5 (2004)]; bulk h-BN: 400 W/m • K [*Progress in Materials Science* **100**, 170–186 (2019)]).

To evaluate the temperature gradient induced by the on-chip heater, we carry out finite element analysis. The device setup is shown in Fig. R8a. The thickness of ultrathin BSCCO is set to be 0.01 (arb. units), which is far less than that of h-BN (0.1) and SiO₂ (0.3). Figure R8b illustrates the temperature gradient caused by a heat flux from the heater (Au).

Fig. R8 a, Cross sectional view of the setup considered for finite element analysis. Thicknesses of BSCCO, Au and h-BN are set to be 0.01, 0.1 and 0.1, respectively. **b**, Temperature gradient induced by a heat flux from the heater. The thermal conductivities of gold strip (Au), BSCCO, h-BN, SiO₂ and Si are set to be 317, 2, 400, 0.5 and 150 W/m • K, respectively. The outer boundary is kept at a constant temperature $T_b = 1$ K. The internal heat source is set to be 10 W/m³ and the heat flux from the bottom of the gold strip is set to be 10 W/m². Yellow rectangles mark the boundary of different materials. Black curves denote the isotherms.

In summary, I think this paper has great potential, but the presentation and high-level analysis does need to be improved.

[Our reply] We would like to thank again the reviewer for the positive remark of our work. We have followed the reviewer's comments and substantially revised our manuscript.

REVIEWER COMMENTS

Reviewer #1 (Remarks to the Author):

In the revised manuscript, the authors emphasize their three key findings ((i)-(iii)) on page 4 in the main text: "A comparative study of ultrathin and bulk-like samples clearly reveals **the dimensionality effect in thermal activation of vortices (\Rightarrow (i))**. By combing the magnetoresistance and Nernst effect data from the same batch of samples, we obtain the vortex sheet entropy S_d^{sheet} from the optimally doped (OP) to the EUD regime. **Notably, S_d^{sheet} dramatically plummets with decreasing doping, substantially deviating from the value of $k_B \ln 2$ (\Rightarrow (ii)). Above T_c , the Nernst data demarcates the boundary for superconducting fluctuations, which only extends to about 40 K above T_c (\Rightarrow (iii))."** However, as I will mention in more detail below ([1]-[3]), these findings do not meet the high acceptance criteria of Nature Communications at this point. I partly acknowledge the novelty of the manuscript in the following points.

- The improved homogeneity of thin films allows the authors to obtain reliable data of ρ_{xx} and N/ρ_{xx} ($\propto S_d$) in UD and EUD regime.
- Among samples with drastically different doping levels, the peak values of $N(\propto \rho_{xx} S_d)$ are on the same order of magnitude, while those of S_d decrease with decreasing doping, which is consistent with Ref. 22.
- The value of λ is extracted from resistivity data and then the relation $S_d \propto \exp(n_s)$ is found. However, there are a number of points that have to be addressed before this article can be considered for publication.

[1] (For (i)) On page 7, the authors find the 2D collective creep of vortices in ultrathin BSCCO sample. However, it has been commonly observed in 2D superconducting films. If the authors found such a behavior in bulk BSCCO sample, it would be interesting.

[2] (For (ii)) I think that the author's finding, $S_d^{\text{sheet}} \propto \exp(n_s)$, is important because a quantitative understanding of the vortex entropy is still lacking. However, the value of $k_B \ln 2$ does not seem to be a universal lower bound of the vortex entropy as the reviewer #3 mentioned. Actually, the author's finding indicates that S_d^{sheet} basically decreases with decreasing n_s . n_s can decrease as a function of temperature and magnetic field as well as a function of doping level. Moreover, it has already been established in theories [Ref.51, Caroli and Maki, Phys. Rev. 164, 591 (1967), etc.] that the vortex entropy decreases to 0 with increasing T or B . These theories have been validated in some experimental literature. Therefore, I believe that the author's finding is variable if the authors withdraw the scenario that they find the violation of the universal lower bound.

Furthermore, the author should explicitly declare the definition of S_d^{sheet} ($\equiv S_d$) on page 11 in the main text. The authors seemingly define S_d^{sheet} as the vortex entropy obtained from the maximum value of $\alpha_{xy}(T)$ ($\equiv N/\rho_{xx} = S_d/\Phi_0$) at fixed B , and regard it as the lowest value of the vortex entropy at fixed B in each sample. Indeed, the authors mention in the reply to the reviewer #3, "Taking the value of N/ρ at the peak position of N may therefore mark a lower bound." However, I do not agree with their statement because the vortex contribution to the entropy further decreases with increasing T or B . The value of S_d^{sheet} obtained from the peak value of $\alpha_{xy}(T)$ may be useful as a representative value for samples with different doping levels.

[3] (For (iii)) On page 4, the authors mention, "Theoretical developments on Gaussian superconducting fluctuations further elucidate the nature of Nernst effect above T_c , showing quantitative agreement with experimental data from superconductors of $\text{Nb}_x\text{Si}_{1-x}$, $\text{La}_{1.8-x}\text{Eu}_{0.2}\text{Sr}_x\text{CuO}_4$ (Eu-LSCO), $\text{Pr}_{2-x}\text{Ce}_x\text{CuO}_4$, etc. Still, data on BSCCO remained limited and the underdoped regime was less explored." This means that the impact of the author's finding is limited. Moreover, in Ref. 30 [Nat. Phys. 8, 751 (2012)], Chang et al. re-analyzed the data of BSCCO (Wang et al., Science 299, 86 (2003)) and reported a similar conclusion to that of the authors. The authors should explicitly mention this previous report.

On page 13, the authors determine T_b as the onset temperature for superconducting fluctuations using a similar protocol of Cyr-Choinière et al. I am a little concerned that the authors think the

superconducting fluctuations completely disappear above T_b , because they mention on page 13, "the temperature regime for superconducting fluctuations lies well within the pseudogap region." Basically, the superconducting fluctuations persist above T_b (but undetectable) as the reviewer #3 also commented. Indeed, in the original paper of Cyr-Choinière et al., they mentioned in the summary, "the latter contribution is only significant in a narrow region of temperature above T_c ."

In addition, the authors should address the following points adequately.

[4] On page 3, the authors mention, "a thermal force that can drive a vortex flow below the superconducting transition temperature (T_c)." However, in 2D superconductors that exhibit the BKT transition, vortices are present between T_c ($\equiv T_{BKT}$) and T_{c0} ($> T_c$), where T_{c0} is the mean-field transition temperature. For the EUD sample S5, if the authors can determine $T_{c0} \sim 20$ K based on some criterion, different trends observed only in sample S5, such as $T_{N,p} \gg T_c$, will be improved. A similar solution can be applied to Supplementary Fig.12 b3.

[5] On page 10, the authors mention that ρ_{xx} is measured in the flux-flow regime. On the other hand, on page 7, they analyze their resistivity data using the thermally activated flux flow (TAFF) form. As I commented previously, the pinning effect is negligible in the flux flow regime, while the pinning is effective in the TAFF regime [See Rev. Mod. Phys. 66, 1125 (1994)]. The TAFF regime corresponds to the linear response regime. So, the discussion on the vortex entropy stays essentially unchanged if the authors use the word TAFF instead of the flux flow.

[6] On page 11 and Method section, the authors did not improve the equation (4). To calculate S_d using the equation (3) given by Sergeev et al. in Ref. 51, the authors take account of the T dependence of $B_c(T)$ ($= B_c(0)\{1 - (T/T_c)^2\}$) and the relation of $B_c(0) = B_{c2}(0)\sqrt{2}\kappa$. However, the authors completely ignore the T dependence of $\xi(T)$, namely the T dependence of $B_{c2}(T)$. This is a contradiction because $B_c(T) = B_{c2}(T)\sqrt{2}\kappa$ (as the author used in the previous version). Why do not the authors use the equation provided by Rischau et al. in the supplementary information of Ref.22? This seems to be derived correctly.

[7] In the discussion on pages 12-13, the authors still claim that a drop of S_d in UD and EUD is attributed to ordering of spin or charge. As I commented previously, if the authors claim this scenario, they should confirm that the vortex entropy takes values close to $k_B \ln 2$ in the overdoped regime, instead of showing a decrease, or should show the correlation between the vortex entropy and the degree of order in the charge and spin ordered states. By considering the author's finding, $S_d^{\text{sheet}} \propto \exp(n_s)$, the above scenario is not necessary.

[8] On page 13, the authors discuss T_b . There, Supplementary Fig.11 c1-5 play an important role. So, plotting T_b in Fig3 or transferring Supplementary Fig.11 c1-5 to Fig.3 would be helpful.

[9] On page 16 in the supplementary information, the authors analyze the flux-flow resistivity. However, resistivity data they obtained is in the TAFF regime not in the flux-flow regime. Thus, their analysis is not valid.

[10] In Supplementary Fig.11 c3, the T_b line should be shifted from 80 K to 90 K.

Reviewer #2 (Remarks to the Author):

In the revised manuscript, I substantially appreciate that Hu and coauthors have revised the manuscript including some Nernst data and discussions, as well as the title. In this version, the authors again emphasize the vortex entropy in the underdoped regime drops sharply with doping and is far away $k_B \ln 2$. Another point claimed by authors is that they at first discovery a dimensionality effect by measuring MR and Nernst effect on ultrathin BSCCO. Generally, thermal-

transports studies using the on-chip setup in BSSCO superconductors based on Nernst effect near T_c can reflect the information of superconducting fluctuations, entropy and cooper pairs, et.al. As a glance look, the authors reply the referees' questions and add more data and analyses. But the key question about the suppression of the vortex entropy at low T_c is still puzzled. In the under-doped regime, the Nernst signal tends to approach zero, and does show an important peak below T_c associated with the motion of vortices, which is different from the case in ref.[22]. In contract, the peak is observed at above T_c in BSSCO. The authors should explain the reason why the Nernst peak appears above T_c if the superconducting fluctuation dominates in the present sample. In addition, the decreasing of Nernst signal with doping does not mean the suppression of vortex entropy, other factors like spin or charge ordering maybe involve. Meanwhile the resistivity in EUD extracted by this manuscript can be enhanced by impurities, e-e or other scattering. As a result, the entropy drops suddenly although the Nernst signal is comparable in all samples. In other words, the absence of solid evidences in this manuscript can be found that the decreasing entropy in EUD regime is mainly from the motion of vortex below T_c . And no peak of Nernst signal below T_c in BSSCO should be different from the case of Ref.[22].

Note in S5 sample the author claims the T_c of 7.1 K in the text, but in the figure 9 of Supplementary it is confused that the resistivity shows a clear transition around 40 K, indicating a SC transition. The peak of Nernst signal appears near 20 K in figure 3. If the author should take the values of resistivity and Nernst signal near peak? This may mislead the readers ?

Reviewer #4 (Remarks to the Author):

The author presents a study on the transport properties of exfoliated BSSCO, with a particular focus on the Nernst effect.

The experiments seem to have been done with care and the experimental data is interesting.

Although the initial feedback from three referees acknowledged the quality of the experimental work, they expressed doubts about the theoretical analysis and interpretation of the data.

In response to the referees' suggestions, the authors have made revisions and enhancements to their manuscript. However, I maintain that the theoretical analysis remains too speculative.

The authors persist in centering their analysis on the discrepancy of the vortex entropy from $k_B \log(2)$. Given the absence of a theoretical basis for a universal value of the vortex entropy and limited experimental evidence supporting this assertion, their analysis and discussion appear speculative and somewhat tenuous. Notably, their assertion linking ordering in the underdoped regime to reduced entropy is not supported by experimental data.

Therefore, I propose that the paper should steer clear of speculative discussions and be submitted to a specialized journal.

Reviewer #1 (Remarks to the Author):

In the revised manuscript, the authors emphasize their three key findings ((i)-(iii)) on page 4 in the main text: “A comparative study of ultrathin and bulk-like samples clearly reveals the dimensionality effect in thermal activation of vortices (\Rightarrow (i)). By combining the magnetoresistance and Nernst effect data from the same batch of samples, we obtain the vortex sheet entropy S_{dsheet} from the optimally doped (OP) to the EUD regime. Notably, S_{dsheet} dramatically plummets with decreasing doping, substantially deviating from the value of $kB\ln 2$ (\Rightarrow (ii)). Above T_c , the Nernst data demarcates the boundary for superconducting fluctuations, which only extends to about 40 K above T_c (\Rightarrow (iii)).” However, as I will mention in more detail below ([1]-[3]), these findings do not meet the high acceptance criteria of Nature Communications at this point. I partly acknowledge the novelty of the manuscript in the following points.

- The improved homogeneity of thin films allows the authors to obtain reliable data of ρ_{xx} and N/ρ_{xx} ($\propto S_d$) in UD and EUD regime.
- Among samples with drastically different doping levels, the peak values of N ($\propto \rho_{xx} S_d$) are on the same order of magnitude, while those of S_d decrease with decreasing doping, which is consistent with Ref. 22.
- The value of λ is extracted from resistivity data and then the relation $S_d \propto \exp(ns)$ is found.

However, there are a number of points that have to be addressed before this article can be considered for publication.

[Our reply] We would like to thank the reviewer for summarizing the novelty of our work and providing constructive comments. We have followed the comments raised by the reviewer and revised our manuscript. We provide below a point-to-point answer.

[1] (For (i)) On page 7, the authors find the 2D collective creep of vortices in ultrathin BSCCO sample. However, it has been commonly observed in 2D superconducting films. If the authors found such a behavior in bulk BSCCO sample, it would be interesting.

[Our reply] We have systematically studied the vortex dynamics in both ultrathin and bulk-like BSCCO samples. The thermal activation is governed by the 2D collective creeping of vortices only in ultrathin BSCCO. In bulk-like samples (Supplementary Note 1), we find that the thermal activation shows a power law dependence ($U \propto B^{-\alpha}$). To our knowledge, such a dimensionality crossover in high temperature superconductors has not been firmly established. We have revised the related discussion to clarify this point.

In fact, our work points to a unique technique to extract the London penetration depth as well as the superfluid phase stiffness—crucial parameters in understanding the mechanism of high temperature superconductivity [Bozovic *et al.*, Nature **536**, 309 (2016)], by utilizing the 2D collective creep in ultrathin samples. It will be useful in the study of a great variety of high temperature superconductors.

[2] (For (ii)) I think that the author's finding, $S_{d\text{sheet}} \propto \exp(ns)$, is important because a quantitative understanding of the vortex entropy is still lacking. However, the value of $k_B \ln 2$ does not seem to be a universal lower bound of the vortex entropy as the reviewer #3 mentioned. Actually, the author's finding indicates that $S_{d\text{sheet}}$ basically decreases with decreasing ns . ns can decrease as a function of temperature and magnetic field as well as a function of doping level. Moreover, it has already been established in theories [Ref.51, Caroli and Maki, Phys. Rev. 164, 591 (1967), etc.] that the vortex entropy decreases to 0 with increasing T or B . These theories have been validated in some experimental literature. Therefore, I believe that the author's finding is variable if the authors withdraw the scenario that they find the violation of the universal lower bound.

[Our reply] We have taken this suggestion and withdrawn the scenario of a universal lower bound.

Furthermore, the author should explicitly declare the definition of $S_{d\text{sheet}}$ ($\equiv S_d$) on page 11 in the main text. The authors seemingly define $S_{d\text{sheet}}$ as the vortex entropy obtained from the maximum value of $\alpha_{xy}(T)$ ($\equiv N/\rho_{xx} = S_d/\Phi_0$) at fixed B , and regard it as the lowest value of the vortex entropy at fixed B in each sample. Indeed, the authors mention in the reply to the reviewer #3, "Taking the value of N/ρ at the peak position of N may therefore mark a lower bound." However, I do not agree with their statement because the vortex contribution to the entropy further decreases with increasing T or B . The value of $S_{d\text{sheet}}$ obtained from the peak value of $\alpha_{xy}(T)$ may be useful as a representative value for samples with different doping levels.

[Our reply] Following this suggestion, we employ the peak value of $\alpha_{xy}(T)$ to derive the representative values of S_d^{sheet} for samples S1 to S4. For sample S5, the peak of α_{xy} is at a temperature higher than T_c such that the corresponding resistivity may involve contribution from quasi-particles. We therefore employ data at 5 K for S5 to evaluate S_d^{sheet} . This definition has been added to the data analysis section starting from page 11.

[3] (For (iii)) On page 4, the author mention, "Theoretical developments on Gaussian superconducting fluctuations further elucidate the nature of Nernst effect above T_c , showing quantitative agreement with experimental data from superconductors of $\text{Nb}_x\text{Si}_{1-x}$, $\text{La}_{1.8-x}\text{Eu}_{0.2}\text{Sr}_x\text{CuO}_4$ (Eu-LSCO), $\text{Pr}_{2-x}\text{Ce}_x\text{CuO}_4$, etc. Still, data on BSCCO remained limited and the underdoped regime was less explored." This means that the impact of the author's finding is limited. Moreover, in Ref. 30 [Nat. Phys. 8, 751 (2012)], Chang et al. re-analyzed the data of BSCCO (Wang et al., Science 299, 86 (2003)) and reported a similar conclusion to that of the authors. The authors should explicitly mention this previous report.

[Our reply] Here we would like to distinguish two aspects of the superconducting fluctuations: (1) they occur in a temperature window above T_c . (2) According to the theory of Gaussian superconducting fluctuations, the off-diagonal Peltier coefficient has a specific temperature dependence.

About point (1), data across a wide doping range is still lacking. We attach the latest work in this direction (ref.28) in Fig. R1 and compare it with our data in Fig. 4. We have revised the

Fig. R1 Comparison between previously obtained data points and our results for marking the onset of superconducting fluctuations from the Nernst effect. The figure on the left is clipped from ref. 28.

corresponding texts in the introduction to focus on this point.

For the second aspect, researchers have analyzed the data of Nb_xSi_{1-x} , $La_{1.8-x}Eu_{0.2}Sr_xCuO_4$ (Eu-LSCO) and $Pr_{2-x}Ce_xCuO_4$ by using the formula derived from Gaussian superconducting fluctuations. For convenience, we collect their analysis in Fig. R2. Although Chang *et al.* (ref. 30) reanalyzed the Nernst data from $Bi_{2-y}La_ySrCuO_{6+x}$ (Bi-2201), they did not carry out such a quantitative analysis as in Fig. R2. We therefore believe our work is the first one that does a quantitative comparison to the theory of Gaussian superconducting fluctuations in BSSCO (Supplementary Note 7). In order to avoid the confusion caused by the previous introduction, we have moved the description on the work of Nb_xSi_{1-x} , $La_{1.8-x}Eu_{0.2}Sr_xCuO_4$ (Eu-LSCO) and $Pr_{2-x}Ce_xCuO_4$ to the discussion section of our paper.

Fig. R2 Clips of data on the quantitative comparison between experiment and the theory of Gaussian superconducting fluctuations. The figures are from ref. 29-31 in the main text.

On page 13, the authors determine T_b as the onset temperature for superconducting fluctuations using a similar protocol of Cyr-Choinière *et al.* I am a little concerned that the authors think the superconducting fluctuations completely disappear above T_b , because they mention on page 13, “the temperature regime for superconducting fluctuations lies well within the pseudogap region.” Basically, the superconducting fluctuations persist above T_b (but undetectable) as the reviewer #3 also commented. Indeed, in the original paper of Cyr-Choinière *et al.*, they mentioned in the summary, “the latter contribution is only significant in a narrow region of temperature above T_c .”

[Our reply] We thank the reviewer for this thoughtful comment. We have revised our texts to be

more rigorous. We define T_b as the temperature above which the Nernst signal from superconducting fluctuations drops below the measurement noise level.

[4] On page 3, the authors mention, “a thermal force that can drive a vortex flow below the superconducting transition temperature (T_c).” However, in 2D superconductors that exhibit the BKT transition, vortices are present between T_c ($\equiv T_{BKT}$) and T_{c0} ($> T_c$), where T_{c0} is the mean-field transition temperature. For the EUD sample S5, if the authors can determine $T_{c0} \sim 20$ K based on some criterion, different trends observed only in sample S5, such as $T_N, \rho \gg T_c$, will be improved. A similar solution can be applied to Supplementary Fig.12 b3.

[Our reply] We agree with the reviewer. To make the readers aware of this fact, we have added that although $T_{N,p}$ greatly exceeds T_c it remains below the onset temperature for superconductivity. We have also added discussions on the onset temperature point in the data analysis related to Fig. 2. As for the analysis of Gaussian superconducting fluctuations, using a different definition of T_c would not affect the main conclusion that the data and theory agree in a certain temperature window. This discussion was provided before to Reviewer #3 in the first round. For convenience, we show the results in Fig. R3 below.

Fig. R3 **a,b** Nernst coefficient in the zero-field limit, v_0 , divided by the zero-field resistivity ρ_0 for sample S5 as a function of the reduced temperature $t (= T/T_c - 1)$. Here T_c is defined as the point where the resistance drops to 1% (panel **a**) or 50% (panel **b**) of the normal state resistance. Dashed lines are obtained by inserting the extracted coherence length ξ_{ab}^0 (6 and 3 nm for **a** and **b**) into the equation for Gaussian superconducting fluctuations (Eq. S2 in the supplementary information).

[5] On page 10, the authors mention that ρ_{xx} is measured in the flux-flow regime. On the other hand, on page 7, they analyze their resistivity data using the thermally activated flux flow (TAFF) form. As I commented previously, the pinning effect is negligible in the flux flow regime, while the pinning is effective in the TAFF regime [See Rev. Mod. Phys. 66, 1125 (1994)]. The TAFF regime corresponds to the linear response regime. So, the discussion on the vortex entropy stays essentially unchanged if the authors use the word TAFF instead of the flux flow.

[Our reply] We have revised the related description according to this suggestion.

[6] On page 11 and Method section, the authors did not improve the equation (4). To calculate S_d using the equation (3) given by Sergeev et al. in Ref. 51, the authors take account of the T dependence of $B_c(T)$ ($= B_c(0)\{1 - (T/T_c)^2\}$) and the relation of $B_c(0) = B_{c2}(0)\sqrt{2}\kappa$. However, the authors completely ignore the T dependence of $\xi(T)$, namely the T dependence of $B_{c2}(T)$. This is

a contradiction because $B_c(T) = B_{c2}(T)\sqrt{2\kappa}$ (as the author used in the previous version). Why do not the authors use the equation provided by Rischau et al. in the supplementary information of Ref.22? This seems to be derived correctly.

[Our reply] We have revised our derivation based on the equation provided by Rischau *et al.* We wish to point out that it does not affect the main conclusions: (1) the theoretical value is a few times larger than the experimental one at optimal doping; (2) the theoretical value does not change much with doping.

[7] In the discussion on pages 12-13, the authors still claim that a drop of S_d in UD and EUD is attributed to ordering of spin or charge. As I commented previously, if the authors claim this scenario, they should confirm that the vortex entropy takes values close to $k_B \ln 2$ in the overdoped regime, instead of showing a decrease, or should show the correlation between the vortex entropy and the degree of order in the charge and spin ordered states. By considering the author's finding, $S_{d\text{sheet}} \propto \exp(ns)$, the above scenario is not necessary.

[Our reply] Motivated by this comment, we realize that there exists clearly no direct correlation between the monotonic decay of our vortex entropy and the spin or charge ordering in the underdoped regime. We have revised our discussions to highlight this point.

[8] On page 13, the authors discuss T_b . There, Supplementary Fig.11 c1-5 play an important role. So, plotting T_b in Fig3 or transferring Supplementary Fig.11 c1-5 to Fig.3 would be helpful.

[Our reply] We have followed this suggestion and updated Fig. 3 by including both T_b and the resistivity data.

[9] On page 16 in the supplementary information, the authors analyze the flux-flow resistivity. However, resistivity data they obtained is in the TAFF regime not in the flux-flow regime. Thus, their analysis is not valid.

[Our reply] By "flux-flow resistivity", we are actually referring to the resistivity in the TAFF regime. We have corrected this in the revised manuscript and Supplementary Information.

[10] In Supplementary Fig.11 c3, the T_b line should be shifted from 80 K to 90 K.

[Our reply] We thank the reviewer for carefully checking our manuscript. We have corrected this wrong labeling.

The editor has informed me that the reviewer #3 is unable to review the manuscript at this round unfortunately. Therefore, I would like to respond to the author's comment for the reviewer #3. Below are some excerpts from the comments of the reviewer #3 and the authors, which are the most important parts in the comments. I underlined the important sentences.

the reviewer #3's comments

Let me provide a bit of context: In Rischau et al. PRL2021 the authors made two claims. The first is that, despite otherwise extremely different parameters (e.g. gap and T_c , lattice constant etc.), the peak (in B-T space) Nernst signal N in numerous superconductors lies in a range somewhere between 1 and 10 $\mu\text{V}/\text{K}$. Although this claim does not entirely hold up, - there are some reports in organic $\kappa\text{-(BEDT-TTF)}_2\text{X}_2$ samples which rather range in the tens of nV/K (<https://www.nature.com/articles/nature06182> and <https://www.nature.com/articles/srep03390>), although one may question those given other measurements in the same material that report higher values ([https://doi.org/10.1016/0921-4534\(96\)00243-2](https://doi.org/10.1016/0921-4534(96)00243-2)) - it does still seem to be strikingly general. In a later work by Behnia (J. Phys.: Condens. Matter 35 (2023) 074003, also cited - I will refer to it as Behnia JPCM) this value is related to a viscosity-to-entropy density ratio, and this is certainly a topic that deserves further exploration.

The second claim in that work is that when dividing the Nernst signal at its peak by the resistivity at the same field and temperature, ρ , a universal bound of $\ln(2)\text{kB}$ per vortex per layer (in that case the "layer" is taken as one lattice constant in the field direction) is not exceeded (and the values reported in that work seem to hit that limit to two significant digits).

The latter claim had two problems. First of all, if the ratio of N/ρ gives access to the entropy S , why should one consider its value at the peak of N ? Instead, one should consider the peak of N/ρ itself. As one can see for example in work by Li et al (Chin. Phys. B Vol. 29, No. 8 (2020) 087402), also cited by the authors, this ratio can actually be much larger at lower fields than it is at the peak of N . If one wants to make general statements about vortex entropy, there is no reason to limit those studies to the peak of N , which is ultimately an experimental quantity (whereas the ratio of N/ρ gives α_{xy} , which is more intrinsic). Second, there are actually a number of materials where this "universal bound" of $\ln(2)\text{kB}$ is exceeded, and in Behnia JPCM2023 those works are indeed mentioned. The value in most of these is still "around kB " (and there is some uncertainty of what constitutes the length-scale of one layer).

However, importantly, in Niobium-based superconductors, the entropy exceeds 200k_B ! (see section "Historical Nernst data on superconducting niobium and its alloys" and refs 56 and 57 in that paper. So one can clearly say that the "universal bound" for S postulated by Rischau et al. simply didn't hold up to further scrutiny, and the current manuscript submitted here is a further confirmation that $\ln(2)\text{kB}$ is not a valid_upper_bound. As for a_lower_bound, that was not postulated there (although in Behnia JPCM there is a brief speculation, but it is enough to take any data at e.g. higher B-field, see for example Li et al, or other temperatures, to see that this does not hold up experimentally - although it is of course not clear in those other works if some

non-flux-flow resistivity also comes in).

The author's comments

We agree with the reviewer that kBl_n2 is not a valid upper bound. Nevertheless, the scenario of a lower bound, as claimed in Behnia JPCM, seems to be valid in the experimental systems that have been explored by so far (before our work).

The reviewer argued against the lower bound scenario by pointing out N/ρ further decreases at higher fields. However, as the reviewer also recognized, this further decrease, after N reaches the peak, can have a mundane origin. Simply, superconductivity in this regime is already strongly suppressed and the flux-flow situation is not applicable. In Fig. R1, we show the data from our own experiments on NbSe₂. N reaches a peak when $\rho(\rho_{xx})$ is already 80% of the normal state resistance. The sample may turn partially non-superconducting. The drop of Nernst signal at higher magnetic field reflects suppressed vortex contribution. From the work of Li et al. (ref. 21 in the updated manuscript)

N/ρ keeps decreasing with increasing magnetic field and smoothly crosses the position where N itself has a peak (Fig. 2 of ref. 21). Taking the value of N/ρ at the peak position of N may therefore mark a lower bound.

My comments

As I already wrote in the section of my comment, I think the value of kBl_n2 does not seem to be a universal lower bound of the vortex entropy as the reviewer #3 mentioned. Actually, the author's finding indicates that S_{ds} basically decreases with decreasing n_s . n_s can decrease as a function of temperature and magnetic field as well as a function of doping level. If the authors claim the lower bound scenario, they should show the B or T dependence of α_{xy} ($= N/\rho$) and should decompose it into the contributions of vortices, gaussian fluctuations, and quasiparticles using the author's own data. And then, they should confirm whether the vortex contribution show the lowest value at the peak position of N . Of course, the same procedure should be applied to the data of Fig. R1.

[Our reply] We would like to thank the reviewer for carefully checking the remarks from Reviewer #3 and our responses. We would like to point out that the lower bound scenario is not claimed by us. It is a scenario claimed in the recent works of Behnia group. As we stated in the answers to the reviewer's other points, we now remove this scenario from our discussions and focus on our experimental findings in the revised paper.

Reviewer #2 (Remarks to the Author):

In the revised manuscript, I substantially appreciate that Hu and coauthors have revised the manuscript including some Nernst data and discussions, as well as the title. In this version, the authors again emphasize the vortex entropy in the underdoped regime drops sharply with doping and is far away KBLn2. Another point claimed by authors is that they at first discovery a dimensionality effect by measuring MR and Nernst effect on ultrathin BSSCO. Generally, thermal-transport studies using the on-chip setup in BSSCO superconductors based on Nernst effect near T_c can reflect the information of superconducting fluctuations, entropy and cooper pairs, et.al. As a glance look, the authors reply the referees' questions and add more data and analyses.

[Our reply] We thank the reviewer for the positive evaluation on our efforts.

But the key question about the suppression of the vortex entropy at low T_c is still puzzled. In the under-doped regime, the Nernst signal tends to approach zero, and does show an important peak below T_c associated with the motion of vortices, which is different from the case in ref.[22]. In contract, the peak is observed at above T_c in BSSCO. The authors should explain the reason why the Nernst peak appears above T_c if the superconducting fluctuation dominates in the present sample.

[Our reply] The reviewer seemed to mean that the Nernst signal in our experiments on underdoped samples tends to approach zero at temperatures below T_c and shows a peak above T_c , whereas the peak occurs below T_c in ref. [22]. We would like to point out that the Nernst peaks indeed occur below T_c , in agreement with that in ref. [22], for samples S1 to S4. The Nernst peak appears above T_c only in sample S5 in the extremely underdoped case. We emphasize again that we keep using the Nernst data below T_c to evaluate the vortex entropy.

Based on the theory of Gaussian superconducting fluctuations, α_{xy} should increase as T approaches T_c from the high temperature side. The Nernst signal $N = \alpha_{xy}\rho_{xx}$ can show a non-monotonic evolution with temperature, due to the competition between the increasing trend of α_{xy} and the decreasing trend of ρ_{xx} : it increases first with decreasing T when ρ_{xx} shows a gradual drop with decreasing T ; The Nernst signal starts to drop with decreasing T when ρ_{xx} shows a rapid drop in the superconducting transition regime before reaching T_c . In total, the superconducting fluctuations can give rise to a Nernst peak above T_c .

In addition, the decreasing of Nernst signal with doping does not mean the suppression of vortex entropy, other factors like spin or charge ordering maybe involve. Meanwhile the resistivity in EUD extracted by this manuscript can be enhanced by impurities, e-e or other scattering. As a result, the entropy drops suddenly although the Nernst signal is comparable in all samples.

[Our reply] We wish to bring the reviewer's attention to our discussion section, in which we have explicitly considered the effect of spin or charge ordering on the vortex entropy. We argue that the emergent ordering either occurs in a limited doping range or has a non-monotonic doping

dependence in the underdoped regime. It is therefore incompatible with the monotonic dependence of vortex entropy in our experiment. For the possible involvement of impurity scattering, we always choose data below T_c such that impurity or electron-electron scattering becomes negligible.

In other words, the absence of solid evidences in this manuscript can be found that the decreasing entropy in EUD regime is mainly from the motion of vortex below T_c . And no peak of Nernst signal below T_c in BSSCO should be different from the case of Ref.[22].

[Our reply] We are not fully clear about this statement. The reviewer seems to argue that our finding of the decreasing entropy across a wide doping regime is not valid because we observe the Nernst peak above T_c in a single sample (sample S5). We would like to point out that: (1) we obtain the Nernst peak below T_c in samples S1 to S4. (2) The reason for a Nernst peak above T_c in S5 could arise from strong superconducting fluctuations.

Note in S5 sample the author claims the T_c of 7.1 K in the text, but in the figure 9 of Supplementary it is confused that the resistivity shows a clear transition around 40 K, indicating a SC transition. The peak of Nernst signal appears near 20 K in figure 3. If the author should take the values of resistivity and Nernst signal near peak? This may mislead the readers?

[Our reply] In this work, we define T_c as the temperature where the resistivity drops to 1% of normal state resistivity at zero magnetic field, i.e., $T_c^{1\%}(B = 0)$, which is close to the zero-resistance temperature employed for analyzing the superconducting fluctuations. For defining T_c from resistivity data of a superconductor, the criterion of using 50% or 90% of ρ_n is also widely employed, e.g., $T_c^{50\%(90\%)} \sim 20$ (40) K for S5 (see Supplementary Fig. 9e).

In order to evaluate the vortex entropy, we take the values of ρ_{xx} and N below T_c . Otherwise, as the reviewer pointed out, the resistivity may be contributed by quasi-particles. Nevertheless, taking values at the peak for S5 would not affect the general doping dependence of S_d^{sheet} , as shown in Fig. R4.

Fig. R4 Sheet entropy (squares) evaluated from the Nernst peak at the maximum B (9 T for S1; 12 T for S4-S5) in our measurements. For comparison, the entropy data (diamonds) and the exponential fit shown in Fig.4b (main text) are included.

Reviewer #4 (Remarks to the Author):

The author presents a study on the transport properties of exfoliated BSCCO, with a particular focus on the Nernst effect.

The experiments seem to have been done with care and the experimental data is interesting.

[Our reply] We would like to thank the reviewer for the positive evaluation on our work.

Although the initial feedback from three referees acknowledged the quality of the experimental work, they expressed doubts about the theoretical analysis and interpretation of the data.

In response to the referees' suggestions, the authors have made revisions and enhancements to their manuscript. However, I maintain that the theoretical analysis remains too speculative.

The authors persist in centering their analysis on the discrepancy of the vortex entropy from $k_B \log(2)$. Given the absence of a theoretical basis for a universal value of the vortex entropy and limited experimental evidence supporting this assertion, their analysis and discussion appear speculative and somewhat tenuous. Notably, their assertion linking ordering in the underdoped regime to reduced entropy is not supported by experimental data.

Therefore, I propose that the paper should steer clear of speculative discussions and be submitted to a specialized journal.

[Our reply] We thank the reviewer for the thoughtful comments. We have taken the suggestions and revised our manuscript to steer clear of the controversial claim of a universal bound for vortex entropy. We have also excluded the possible involvement of ordering as a mechanism for the reduced entropy. We believe our paper has been substantially strengthened and meets the criteria of Nature Communications.

REVIEWER COMMENTS

Reviewer #1 (Remarks to the Author):

The authors have answered all my comments and revised their manuscript. In their study, the homogeneity of BSCCO samples is improved by using ultrathin films. Although measuring the thermoelectric effect in ultrathin samples is a challenging experiment, the authors obtained reliable data of resistance and Nernst signals in UD and EUD regime. Consequently, the authors found that the vortex entropy S_d decreases with decreasing doping in the form of $S_d \propto \exp(n_s)$. Moreover, the authors demarcated the temperature regime for prominent superconducting fluctuations in the phase diagram. These results provide deeper insight into the nature of the high- T_c superconductors and the vortex entropy. I therefore recommend the manuscript for publication in Nature Communications after the following revision.

On page 11, the authors write, " ρ_{xx} vanishes at zero magnetic field and grows linearly with B across a wide field range (Supplementary Fig. 8). This trend agrees with the thermally activated flux flow behavior: $\rho_{xx} = B\Phi_0/\eta$ for a constant η ." However, this explanation is incorrect; $\rho_{xx} \propto B$ is not a trend of the thermally activated flux flow (TAFF) but that of flux flow (FF). The authors still confuse TAFF and FF. As the authors themselves mention in the manuscript, $\rho_{xx} \propto \exp(-U/k_B T)$ and $U \propto -\ln B$ in the TAFF regime. This confusion seems to be caused by the need to derive $S_d = \Phi_0 a_{xy}$. Even in the TAFF regime, $S_d = \Phi_0 a_{xy}$ seems to be derived from the balance equation between a driving force (a thermal force or a Lorentz-like force), a viscous force, and a pinning force.

On page 4, "By combing the magneto-resistance" -> "By combining the magneto-resistance" In Fig. 4a, a dashed line around k_B (given on page 12) is missing.

Reviewer #2 (Remarks to the Author):

The revised manuscript reports the decreasing vortex entropy in the UD BSCC samples, comparing with other SC systems. It is helpful to understand the superconducting fluctuation in BSCC if the evidence is solid. In addition, In a superconductor with the optimal T_c , the Nernst signal caused by the contribution of flux flow in the vortex liquid state always peaks at around T_c , in which the related resistivity drops substantially but remains a finite value, thus leading to a finite vortex entropy. Whereas, the Nernst signal tends to disappear in the vortex lattice state, where the resistivity is zero. Thus, in the EUD regime, the Nernst signal peaks beyond T_c and can involve some other factors like spin or charge order as well as SC fluctuation. Meanwhile, the resistivity (b3, b4 and b5) is enhanced by other scatterings. Thus I am still afraid that it is very hard to get the true vortex entropy from the motion of vortex in the UD samples.

Reviewer #4 (Remarks to the Author):

As already said in my previous report, the presented experimental work is remarkable and the data on these ultra-thin flakes of BSCCO is interesting. I appreciate that the authors removed the speculative discussion about the universal value of vortex entropy. With a suitable presentation of the state of art and a suitable analysis of these data, the paper should be suitable for Nature Com. Unfortunately, their presentation of our current understanding of superconducting fluctuations in cuprates and the origin of the superconducting dome is awkward and seems to reflect a misunderstanding of the current situation. The consequence is that their presentation of the data is inconsistent as I will detail below, for this reason, I don't think that the paper can be published in Nature Com in the current state.

To understand my comments, a reminder of some historical developments is needed.

In 1989, in ref[45], Uemura proposed that the superconducting critical temperature was controlled

by the superfluid density. Then, Emery and Kivelson extended the concept of phase-coherence temperature introduced by Berezinski, Kosterlitz and Thouless and suggested that, for any superconductor, vortex-antivortex pairs should appear spontaneously when the thermal energy is larger than the energy cost for their formation. This defines a characteristic temperature for phase coherence, TCOH, which value is controlled by the superfluid density. Above this temperature, spontaneous nucleation of vortices is possible. In conventional superconductors, this coherence temperature largely exceeds TBCS, the Cooper pair forming temperature, and superconducting fluctuations exist only as Cooper pair fluctuations (Gaussian fluctuations). In this case, the critical temperature is set by TBCS, which is not controlled by the superfluid density. In contrast, for low density superconductors, TCOH may become smaller than TBCS. This implies that the superconducting transition is controlled by TCOH and so by the superfluid density. This is of the origin of the relation between superfluid density and the superconducting critical temperature that the authors use in the manuscript.

In 2000, in ref[27], ignoring that Gaussian fluctuations above TBCS could produce a Nernst signal, the authors attributed their measured Nernst signal to vortex-like excitations expected between TCOH and TBCS, given that $TCOH \ll TBCS$, consistent with the fashion of the time but which is, according to our current understanding, unlikely.

In 2006, in ref[29], using a conventional disordered superconductor NbSi with short mean free path where the Nernst signal due to quasiparticle is negligible, the Nernst signal due to Gaussian fluctuations above TBCS was clearly observed. This is a solid result because the data can be compared quantitatively with the BCS theory with no adjustable parameters. I remind that the BCS theory is probably one of the most solid and successful theory of condensed-matter physics. Furthermore, note that for Gaussian fluctuations, there is no known upper limit for the existence of superconducting fluctuations. Only experimental resolution and the capacity to discriminate between the Nernst contributions arising from either superconducting fluctuations or quasiparticles provides the upper temperature. In ref[29], Gaussian fluctuations are observed up to 30 times T_c .

In 2012, in ref[30,31], the Gaussian fluctuations were observed above T_c in the cuprates. The caption of Ref[30] close with the sentence "We conclude that competing states such as stripe order weaken superconductivity and this, rather than phase fluctuations, causes T_c to fall as cuprates become underdoped.". This means that there is no such things as vortex-like fluctuations above T_c in the cuprates, the superconducting critical temperature is not controlled by the superfluid density and the measured critical temperature is TBCS and not TCOH. Furthermore, the Nernst signal above T_c is controlled by Gaussian fluctuations in cuprates as shown in refs[9,30,31].

Following this historical account, I hope that the authors understand that, for the same samples, saying that the superconducting critical temperature is controlled by the superfluid density and the fluctuations are described by Gaussian fluctuations does not make sense.

If T_c is controlled by the superfluid density, then fluctuations should be due to vortex-like fluctuations (not Gaussian). On the other hand, if superconducting fluctuations are Gaussian (which I believe is your most solid result of the data), then the critical temperature is controlled by TBCS, which is not controlled by superfluid density.

To progress on the writing of this article, I suggest having a deeper look at the following questions:
a) The use of the Uemura formula is based on the observation of the linearity on the semilog plot fig2h. The U range of this plot is very small, making the distinction between a semilog plot and a loglog plot not obvious. The authors should show, at least, a plot showing the ultra-thin films data on a log-log plot, to allow a more direct comparison with Fig.S4b where the data for bulk crystal are shown on a log-log plot.

Following this comment, I believe that the statement line 158- « Our work therefore reveals clearly the dimensionality crossover in high temperature cuprate superconductors from multi-layer to the atomic limit. » is not convincing.

b) What made the identification of Gaussian fluctuations in NbSi[29] possible is that the quasiparticle contribution was negligible. Generally, the quasiparticle contribution is expected to be proportional to the mobility or the mean free path. In cuprates[30,31], the difficulty is to separate the quasiparticle contribution from the superconducting fluctuations. However, in your

data, it seems that the quasiparticle contribution is small. This may be due to short mean free path in these ultra-thin samples in comparison to bulk crystals. You should provide more details on the mean free path in these ultra-thin samples. Then, you should estimate the quasiparticle contribution by multiplying the Seebeck coefficient by the Hall angle, see ref[29] for example. Thus, you could discuss in more details the quasiparticle contributions and Gaussian fluctuations contributions. Actually, with the sample fabrication, I believe that your observation that the Nernst signal is controlled by Gaussian fluctuations, without (it seems) much quasiparticle contribution, is the most interesting part of the data. I would focus on this.

Reviewer #1 (Remarks to the Author):

The authors have answered all my comments and revised their manuscript. In their study, the homogeneity of BSCCO samples is improved by using ultrathin films. Although measuring the thermoelectric effect in ultrathin samples is a challenging experiment, the authors obtained reliable data of resistance and Nernst signals in UD and EUD regime. Consequently, the authors found that the vortex entropy S_d decreases with decreasing doping in the form of $S_d \propto \exp(ns)$. Moreover, the authors demarcated the temperature regime for prominent superconducting fluctuations in the phase diagram. These results provide deeper insight into the nature of the high- T_c superconductors and the vortex entropy. I therefore recommend the manuscript for publication in Nature Communications after the following revision.

[Our reply] We would like to thank the reviewer for recommending our paper for publication.

On page 11, the authors write, “ ρ_{xx} vanishes at zero magnetic field and grows linearly with B across a wide field range (Supplementary Fig. 8). This trend agrees with the thermally activated flux flow behavior: $\rho_{xx} = B\Phi_0/\eta$ for a constant η .” However, this explanation is incorrect; $\rho_{xx} \propto B$ is not a trend of the thermally activated flux flow (TAFF) but that of flux flow (FF). The authors still confuse TAFF and FF. As the authors themselves mention in the manuscript, $\rho_{xx} \propto \exp(-U/k_B T)$ and $U \propto -\ln B$ in the TAFF regime. This confusion seems to be caused by the need to derive $S_d = \Phi_0 \alpha xy$. Even in the TAFF regime, $S_d = \Phi_0 \alpha xy$ seems to be derived from the balance equation between a driving force (a thermal force or a Lorentz-like force), a viscous force, and a pinning force.

[Our reply] The reviewer is correct that $\rho_{xx} \propto B$ indicates flux flow (FF) whereas $\rho_{xx} \propto \exp(-U/k_B T)$ reflects thermally activated flux flow (TAFF). We have corrected the statements in the revised manuscript.

The reviewer raised a very good point in the first round that FF and TAFF are two distinct regimes but our data seem to show both $\rho_{xx} \propto \exp(-U/k_B T)$, when we try to extract the superfluid stiffness, and $\rho_{xx} \propto B$, when we try to extract the vortex entropy. We now fully appreciate this seeming contradiction. Below, we provide an explanation:

As pointed out by the reviewer and stated clearly in [Rev. Mod. Phys. 66, 1125 (1994)], TAFF occurs typically at relatively lower temperatures and magnetic fields and it crosses over to FF by raising the temperature and magnetic fields. In fact, the thermally activated behavior of $\rho_{xx} \propto \exp(-U/k_B T)$ best describes our data at low temperatures (as can be seen in Fig. R1a below), whereas we show $\rho_{xx} \propto B$ at higher temperatures, i.e., when the system crosses over to the FF regime. For example, we show in Fig. R1b that the Nernst peak we obtained is at a temperature (dashed line) well above the upper boundary for the TAFF regime.

Fig. R1 **a**, Arrhenius plot of the temperature dependent resistivity ρ_{xx} for ultrathin BSCCO sample S1. Dashed lines are linear fits. Circles mark the position where ρ_{xx} deviates from the linear fits by 10%. They represent the upper bound for the thermally activated flux flow regime. **b**, Comparison of the TAFF regime and the peak position of the Nernst signal in S1. The solid line is a guide to eyes. The dashed line marks the temperature at the peak position of Nernst signals at 9 T.

We have added some discussions on this distinction between TAFF and FF in the paragraph introducing the $\rho_{xx} \propto B$ relation. Also, we have revised our data analysis on extracting the vortex entropy in Fig. 4a by using the Nernst peak signal at the highest magnetic field (for sample S5, we still use data at 5 K, at which FF dominates), in order to stay as far away as possible from the TAFF regime.

On page 4, “By combing the magneto-resistance” -> “By combining the magneto-resistance”

[Our reply] We have corrected this typo.

In Fig. 4a, a dashed line around kB (given on page 12) is missing.

[Our reply] We have added the dashed line at k_B in the revised Fig. 4a.

Reviewer #2 (Remarks to the Author):

The revised manuscript reports the decreasing vortex entropy in the UD BSCCO samples, comparing with other SC systems. It is helpful to understand the superconducting fluctuation in BSCC if the evidence is solid. In addition, in a superconductor with the optimal T_c , the Nernst signal caused by the contribution of flux flow in the vortex liquid state always peaks at around T_c , in which the related resistivity drops substantially but remains a finite value, thus leading to a finite vortex entropy. Whereas, the Nernst signal tends to disappear in the vortex lattice state, where the resistivity is zero. Thus, in the EUD regime, the Nernst signal peaks beyond T_c and can involve some other factors like spin or charge order as well as SC fluctuation. Meanwhile, the resistivity (b3, b4 and b5) is enhanced by other scatterings. Thus I am still afraid that it is very hard to get the true vortex entropy from the motion of vortex in the UD samples.

[Our reply] We would like to thank the reviewer for the thoughtful comments. Indeed, Samples S1 to S4 have their Nernst peaks around T_c but sample S5 shows the Nernst peak beyond T_c . For this particular sample—S5, the successful fitting of our data by using the Gaussian superconducting fluctuations (GSF) seems to indicate that spin or charge order is not playing a dominant role. This by no means suggests that spin or charge order can be fully excluded by our experimental results. We have added the highlighted description to the discussion on GSF in the main text: “There exists reasonable agreement in a temperature window above T_c , **indicating a dominant contribution of GSF over other factors such as quasi-particles, spin or charge order.**”

For the enhanced resistivity from S1, S2 to S3-S5, we would like to point out that it does not necessarily indicate the involvement of additional scattering sources. One has to take into account the reduction in carrier densities. A better way to investigate the possible scattering, in the normal state, is to extract the mean free path of the sample, as shown in Fig. R2 below. We point out that the mean free path decreases gradually with reduced doping. This is reasonable because the screening of impurities becomes weaker at a lower carrier density, giving rise to slightly enhanced scattering. Also, there is a dip of mean free path for sample S4, possibly due to sample-to-sample variation. Still, the extracted vortex entropy shows a monotonic decay from S1 to S5. It indicates that the superconducting properties are not directly related to the normal state transport.

Fig. R2 Estimated mean free path l of samples S1 to S5.

Reviewer #4 (Remarks to the Author):

As already said in my previous report, the presented experimental work is remarkable and the data on these ultra-thin flakes of BSCCO is interesting. I appreciate that the authors removed the speculative discussion about the universal value of vortex entropy. With a suitable presentation of the state of art and a suitable analysis of these data, the paper should be suitable for Nature Com.

[Our reply] We would like to thank the reviewer for the very positive remark on our experimental efforts.

Unfortunately, their presentation of our current understanding of superconducting fluctuations in cuprates and the origin of the superconducting dome is awkward and seems to reflect a misunderstanding of the current situation. The consequence is that their presentation of the data is inconsistent as I will detail below, for this reason, I don't think that the paper can be published in Nature Com in the current state.

To understand my comments, a reminder of some historical developments is needed.

In 1989, in ref[45], Uemura proposed that the superconducting critical temperature was controlled by the superfluid density. Then, Emery and Kivelson extended the concept of phase-coherence temperature introduced by Berezinski, Kosterlitz and Thouless and suggested that, for any superconductor, vortex-antivortex pairs should appear spontaneously when the thermal energy is larger than the energy cost for their formation. This defines a characteristic temperature for phase coherence, TCOH, which value is controlled by the superfluid density. Above this temperature, spontaneous nucleation of vortices is possible. In conventional superconductors, this coherence temperature largely exceeds TBCS, the Cooper pair forming temperature, and superconducting fluctuations exist only as Cooper pair fluctuations (Gaussian fluctuations). In this case, the critical temperature is set by TBCS, which is not controlled by the superfluid density. In contrast, for low density superconductors, TCOH may become smaller than TBCS. This implies that the superconducting transition is controlled by TCOH and so by the superfluid density. This is of the origin of the relation between superfluid density and the superconducting critical temperature that the authors use in the manuscript.

In 2000, in ref[27], ignoring that Gaussian fluctuations above TBCS could produce a Nernst signal, the authors attributed their measured Nernst signal to vortex-like excitations expected between TCOH and TBCS, given that $TCOH \ll TBCS$, consistent with the fashion of the time but which is, according to our current understanding, unlikely.

In 2006, in ref[29], using a conventional disordered superconductor NbSi with short mean free path where the Nernst signal due to quasiparticle is negligible, the Nernst signal due to Gaussian fluctuations above TBCS was clearly observed. This is a solid result because the data can be compared quantitatively with the BCS theory with no adjustable parameters. I remind that the BCS theory is probably one of the most solid and successful theory of condensed-matter physics.

Furthermore, note that for Gaussian fluctuations, there is no known upper limit for the existence of superconducting fluctuations. Only experimental resolution and the capacity to discriminate between the Nernst contributions arising from either superconducting fluctuations or quasiparticles provides the upper temperature. In ref[29], Gaussian fluctuations are observed up to 30 times T_c .

In 2012, in ref[30,31], the Gaussian fluctuations were observed above T_c in the cuprates. The caption of Ref[30] close with the sentence “We conclude that competing states such as stripe order weaken superconductivity and this, rather than phase fluctuations, causes T_c to fall as cuprates become underdoped.”. This means that there is no such things as vortex-like fluctuations above T_c in the cuprates, the superconducting critical temperature is not controlled by the superfluid density and the measured critical temperature is TBCS and not TCOH. Furthermore, the Nernst signal above T_c is controlled by Gaussian fluctuations in cuprates as shown in refs[9,30,31].

Following this historical account, I hope that the authors understand that, for the same samples, saying that the superconducting critical temperature is controlled by the superfluid density and the fluctuations are described by Gaussian fluctuations does not make sense.

If T_c is controlled by the superfluid density, then fluctuations should be due to vortex-like fluctuations (not Gaussian). On the other hand, if superconducting fluctuations are Gaussian (which I believe is your most solid result of the data), then the critical temperature is controlled by TBCS, which is not controlled by superfluid density.

[Our reply] We would like to thank the reviewer for the kind introduction of the background and we now recognize the seeming contradiction. However, experimentally we can indeed extract a linear correlation between superfluid density and T_c , as we will show in the next point. This correlation does not necessarily mean that T_c is controlled by the superfluid density. We have revised the corresponding descriptions throughout the manuscript to make this point clear. Specifically, instead of using $\rho_s \propto n_s \propto T_c$, we state that: “our thermal activation study indicates that the superfluid density n_s or ρ_s has a linear relation with T_c .”

As the reviewer pointed out, the scenario of phase fluctuations proposed by Emery and Kivelson is a theoretical interpretation, while the Uemura relation is an experimental observation. The theory of phase fluctuations may be incompatible with the theory for Gaussian superconducting fluctuations but this incompatibility does not invalidate the experimentally observed linear relation.

To progress on the writing of this article, I suggest having a deeper look at the following questions:
a) The use of the Uemura formula is based on the observation of the linearity on the semilog plot fig2h. The U range of this plot is very small, making the distinction between a semilog plot and a loglog plot not obvious. The authors should show, at least, a plot showing the ultra-thin films data on a log-log plot, to allow a more direct comparison with Fig.S4b where the data for bulk crystal are shown on a log-log plot.

Following this comment, I believe that the statement line 158- « Our work therefore reveals clearly the dimensionality crossover in high temperature cuprate superconductors from multi-

layer to the atomic limit. » is not convincing.

[Our reply] Following this suggestion, we show in Fig. R3 (which has been added to the supplementary information) the comparison by using both log-log and semi-log plot. For ultrathin BSCCO samples, $U(B)$ follows a rather linear dependence in the semi-log plot (Fig. R3a) but is obviously curved in the log-log plot (Fig. R3b). By contrast, $U(B)$ in bulk-like BSCCO samples is nonlinear in a semi-log plot (Fig. R3c) but exhibits a linear behavior in the log-log plot (Fig. R3d). It is based on this sharp contrast that we claim there is a clear dimensionality crossover.

Fig. R3 **a,b**, Activation energy U normalized by k_B as a function of B for ultrathin samples in a semi-log plot (**a**) and a log-log plot (**b**). **c,d**, U/k_B vs. B for bulk-like samples in a semi-log plot (**c**) and a log-log plot (**d**). The solid lines in panel **a** and **d** are linear fits. The dashed lines in panel **b** and **c** are guides to the eye.

b) What made the identification of Gaussian fluctuations in NbSi[29] possible is that the quasiparticle contribution was negligible. Generally, the quasiparticle contribution is expected to be proportional to the mobility or the mean free path. In cuprates[30,31], the difficulty is to separate the quasiparticle contribution from the superconducting fluctuations. However, in your data, it seems that the quasiparticle contribution is small. This may be due to short mean free path in these ultra-thin samples in comparison to bulk crystals. You should provide more details on the mean free path in these ultra-thin samples. Then, you should estimate the quasiparticle contribution by multiplying the Seebeck coefficient by the Hall angle, see ref[29] for example. Thus, you could discuss in more details the quasiparticle contributions and Gaussian fluctuations contributions. Actually, with the sample fabrication, I believe that your observation that the Nernst signal is controlled by Gaussian fluctuations, without (it seems) much quasiparticle contribution, is the most interesting part of the data. I would focus on this.

[Our reply] First, the mean free path l for both ultrathin and bulk-like BSCCO samples are summarized in Fig. R4 below. Indeed, the mean free path of ultrathin BSCCO is shorter than that

of the bulk-like BSCCO with comparable doping. We also include the mean free path of $\text{Pr}_{2-x}\text{Ce}_x\text{CuO}_4$ (PCCO) from ref. [31] that the reviewer pointed out. The mean free paths of samples S3-S5 are much shorter than that of PCCO such that the contribution from quasi-particles can be negligible. We note that data from these three samples are employed for the analysis of Gaussian superconducting fluctuations.

Fig. R4 Mean free path l in ultrathin, bulk-like BSCCO and bulk PCCO samples.

Secondly, we estimate the magnitude of $S \cdot \tan\theta/B$ (here the Hall angle is obtained at the normal state) and compare it with the Nernst coefficient ν_0 in the regime of superconducting fluctuations. Table R1 shows this comparison. Clearly, the former value is about two orders of magnitude smaller than the latter one.

Both points—mean free path and $S \cdot \tan\theta/B$ —are now added to Supplementary Note 7 for a more comprehensive discussion on the Gaussian superconducting fluctuations.

Table R1. Magnitude of $\text{Stan}\theta/B$ and the Nernst coefficient ν_0 in the zero-field limit for sample S3-S5. $\text{Stan}\theta/B$ is evaluated at 100 K and 1 T. The values of ν_0 are taken in the temperature window where ν_0/ρ_0 shows a $1/t$ dependence as shown in Supplementary Fig. 12.

	$\frac{\text{Stan}\theta}{B}$ (nV/K T)	ν_0 (nV/K T)
S3	3.04	200-500
S4	5.87	150-400
S5	6.47	200-900

REVIEWERS' COMMENTS

Reviewer #1 (Remarks to the Author):

The authors replied my previous comments. I therefore recommend the manuscript for publication in Nature Communications.

Reviewer #2 (Remarks to the Author):

The authors have revised the manuscript and also answered the comments concerned by the referee. The study of vertex entropy S_d in the ultrathin BSCCO is important to understand the origin of High-Tc SC. These results are interesting and can be published in the journal.

Reviewer #1 (Remarks to the Author):

The authors replied my previous comments. I therefore recommend the manuscript for publication in Nature Communications.

[Our reply] We thank the reviewer for recommending our paper for publication.

Reviewer #2 (Remarks to the Author):

The authors have revised the manuscript and also answered the comments concerned by the referee. The study of vertex entropy S_d in the ultrathin BSCCO is important to understand the origin of High-Tc SC. These results are interesting and can be published in the journal.

[Our reply] We thank the reviewer for the positive evaluation on our work and recommending our work for publication.

For your information, Reviewer #4 was also supportive of publication in private comments to the Editor.

[Our reply] We thank the editor for providing the good news and the reviewer for consent to publication.